# Unifying Approaches in Active Learning and Active Sampling via Fisher Information and Information-Theoretic Quantities

**Andreas Kirsch**                                                    *andreas.kirsch@cs.ox.ac.uk*
**Yarin Gal**                                                              *yarin.gal@cs.ox.ac.uk*
*OATML, Department of Computer Science*
*University of Oxford*

**Reviewed on OpenReview:** *https: // openreview. net/ forum? id= UVDAKQANOW*

## Abstract

Recently proposed methods in data subset selection, that is active learning and active sampling, use Fisher information, Hessians, similarity matrices based on gradients, and gradient lengths to estimate how informative data is for a model's training. Are these different approaches connected, and if so, how? We revisit the fundamentals of Bayesian optimal experiment design and show that these recently proposed methods can be understood as approximations to information-theoretic quantities: among them, the mutual information between predictions and model parameters, known as *expected information gain* or BALD in machine learning, and the mutual information between predictions of acquisition candidates and test samples, known as *expected predictive information gain*. We develop a comprehensive set of approximations using Fisher information and observed information and derive a unified framework that connects seemingly disparate literature. Although Bayesian methods are often seen as separate from non-Bayesian ones, the sometimes fuzzy notion of "informativeness" expressed in various non-Bayesian objectives leads to the same couple of information quantities, which were, in principle, already known by Lindley (1956) and MacKay (1992).

## 1 Introduction

Label and training efficiency are key to a wider deployment of deep learning. Deep learning generally requires a lot of data, much of which must be annotated. This is expensive and time-consuming. Together with semisupervised and unsupervised approaches, *active learning* (Atlas et al., 1990; Cohn et al., 1994) helps increase label efficiency: given access to unlabeled data, active learning selects the most informative samples to label for a given model, thus decreasing the number of required annotations to reach a given level of performance. In addition to label efficiency, training deep learning models is also expensive and time-consuming, and *active sampling* improves training efficiency by filtering the training set to focus on the samples that will be the most informative for the model.

Several new approaches in data subset selection for deep learning have been proposed recently: amongst them, BADGE (Ash et al., 2019), BAIT (Ash et al., 2021), PRISM[1](Kothawade et al., 2022), SIMILAR[1] (Kothawade et al., 2021), and GraNd (Paul et al., 2021). The acquisition functions used to select informative samples in these approaches can be traced back to information-theoretic quantities (short: *information quantities*) that are known from Bayesian optimal experiment design (Lindley, 1956; MacKay, 1992). This connection is the topic of this work.

By examining how Fisher information and second-order posterior approximations (Gaussian approximations) can be used for estimating information quantities, we develop a unifying perspective and relate these recent methods to information quantities used in Bayesian active learning: for active learning, the *expected information gain (EIG)*, also known as (Batch-)BALD (Lindley, 1956; Houlsby et al., 2011; Kirsch et al., 2019), the *(joint) expected predictive information gain (JEPIG or EPIG, respectively)* (Kirsch et al., 2021b;

MacKay, 1992), and, for active sampling, the *information gain (IG)* (Sun et al., 2022) and *(joint) predictive information gain (JPIG or PIG, respectively)* (Mindermann et al., 2022).

These connections point towards possible failure modes of current methods and potential extensions in principled ways. We examine approximations that lead to last-layer approaches, find a potential bias when using similarity matrices, compare trace and log determinant approximations in regard to batch acquisition pathologies, and trade off weight- and prediction-space methods in principle.

**Outline.** To achieve this, we look at second-order posterior approximations (Gaussian approximations) in §3, which we use to revisit Fisher information, its properties, special cases, and approximations in §4. Our contribution here is to summarize results and provide a consistent notation that simplifies reasoning about information quantities, observed information, and Fisher information.

In §5, we approximate the information quantities mentioned above using observed information and Fisher information. We provide a comprehensive overview to understand the differences and similarities and make it easier to spot applications of these approximations in the literature. We pay special attention to the limitations: for example, we will see that some approximations that use the trace of the Fisher information do not take redundancies between samples into account. They exhibit the same pathologies as other methods that, in essence, score points individually, also known as top-k batch acquisition (Kirsch et al., 2021a). In §6, we expand our approach to approximations that use similarity matrices of log-loss gradients. Our contributions are a comprehensive overview of the approximations and the connection to similarity matrices.

In §7, we show that (Batch-)BALD and EPIG on the one hand; and BADGE, BAIT, PRISM[1], and SIMILAR[1] on the other hand can be seen as optimizing the same objectives. The difference is that (Batch-)BALD (Houlsby et al., 2011; Kirsch et al., 2019) and EPIG (Kirsch et al., 2021b) operate in prediction space, while Fisher information-based methods operate in weight space: we show that an approximation of EPIG, a transductive active learning objective, using Fisher information, matches the BAIT objective (Ash et al., 2021). Similarly, we show how BADGE (Ash et al., 2019) approximates the EIG, using the connection to similarity matrices. Finally, we find that *submodularity*-based approaches (Iyer et al., 2021) such as SIMILAR (Kothawade et al., 2021) and PRISM (Kothawade et al., 2022), which report their best results using the log determinant of similarity matrices, approximate information quantities when they perform best. We also show that gradient-length-based methods like EGL (Settles et al., 2007) and GraNd (Paul et al., 2021) can be connected to information quantities.

**Limitations.** Although our results employ a hierarchy of approximations, we do not examine the error terms in detail. This is in line with how these approximations are used in deep learning, where the approximations often only provide motivation for useful mechanisms. However, we try to identify where these approximations might break, enumerate their limitations, and raise several (empirical) research questions for future work.

## 2 Background & Setting

This section introduces the relevant notation, concepts, and probabilistic model that we use in this paper.

**Information Theory.** We follow the practical notation of Kirsch and Gal (2021). In particular, for entropy, we use an implicit or explicit notation, $H[X]$ or $H(p(X))$, while $H(p \parallel q)$ denotes the cross-entropy similar to the Kullback-Leibler divergence $D_{KL}(p \parallel q)$, and $H(p(x))$ denotes Shannon's information content[2]:

$$H(p(x)) := -\log p(x), \tag{1}$$

$$H[x] := H(p(x)), \tag{2}$$

$$H(p(X) \parallel q(X)) := \mathbb{E}_{p(x)}[H(q(x))], \tag{3}$$

$$H[X] := H(p(X)) := H(p(X) \parallel p(X)), \tag{4}$$

---

[1] using log determinant objectives

[2] In related literature, Shannon's information content is often written as $h(p(x))$. We use a unified notation as Shannon's information content is but a point-wise entropy, and we can differentiate between the two by whether we compute it for an outcome $x$ or a random variable $X$ in expectation; see Kirsch and Gal (2021).

where p and q are probabilities distributions, $X$ is a random variable, and $x$ is an outcome. Conditional and joint entropies are defined as usual (note that we also take an expectation over $y$):

$$\mathrm{H}[X \mid Y] = \mathbb{E}_{\mathrm{p}(x,y)}[-\log \mathrm{p}(x \mid y)]. \tag{5}$$

The mutual information $\mathrm{I}[X;Y]$ for random variables $X$ and $Y$ is defined as:

$$\mathrm{I}[X;Y] := \mathrm{H}[X] - \mathrm{H}[X \mid Y], \tag{6}$$

and is also called expected uncertainty reduction or *expected information gain* (Lindley, 1956) because $\mathrm{H}[X]$ quantifies the uncertainty about $X$, and $\mathrm{H}[X \mid Y]$ about $X$ after observing $Y$ (in expectation).

Furthermore, when we mix random variables with outcomes of random variables in (conditional) entropies, for example, $\mathrm{H}[X, Y = y \mid Z]$, or short $\mathrm{H}[X, y \mid Z]$, we expand this to an expectation over the random variables conditional on the outcomes:

$$\mathrm{H}[X, y \mid Z] = \mathbb{E}_{\mathrm{p}(x,z|y)}[-\log \mathrm{p}(x, y \mid z)]. \tag{7}$$

**Probabilistic Model.** We assume a supervised setting: for inputs $X$, we have a Bayesian model with parameters $\Omega$ that makes predictions $Y$. What makes the model Bayesian is that the parameters follow a distribution $\mathrm{p}(\omega)$. We use the probabilistic model:

$$\mathrm{p}(y, \omega \mid x) = \mathrm{p}(y \mid x, \omega) \, \mathrm{p}(\omega). \tag{8}$$

We extend this model to additional data $\mathcal{D} := \{(x_i, y_i)\}$ as follows:

$$\mathrm{p}(\{y_i\}, \omega \mid \{x_i\}) = \mathrm{p}(\{y_i\} \mid \{x_i\}, \omega) \, \mathrm{p}(\omega) = \mathrm{p}(y_1, \ldots, y_n \mid x_1, \ldots, x_n, \omega) \, \mathrm{p}(\omega). \tag{9}$$

That is, we examine the common discriminative case where, unlike in the generative case, we do not model $\mathrm{p}(x)$. The corresponding marginal prediction of the model is $\mathrm{p}(y \mid x) = \mathbb{E}_{\mathrm{p}(\omega)}[\mathrm{p}(y \mid x, \omega)]$.

**Transductive Objectives.** When an objective uses (additional) data $\mathcal{D}^{\mathrm{eval}}$, unlabeled or labeled, to guide acquisitions, we refer to the objective as a *transductive* objective (Yu et al., 2006; Wang et al., 2020).

**Active Learning.** To increase label efficiency, instead of labeling data indiscriminately, active learning iteratively selects and acquires labels for the *most informative* unlabeled data from a *pool set* $\mathcal{D}^{\mathrm{pool}}$ according to some *acquisition function*. An acquisition function scores the informativeness of an unlabeled candidate sample $x^{\mathrm{acq}}$, and the sample that maximizes this score is selected for labeling. After each acquisition step, the model is retrained to take the newly labeled data into account. Labels can be acquired individually or in batches (*batch acquisition*, see below). The *expected information gain (EIG)*

$$\mathrm{I}[\Omega; Y^{\mathrm{acq}} \mid x^{\mathrm{acq}}] \tag{EIG/BALD}$$

and *(joint) expected predictive information gain (JEPIG and EPIG, respectively)*

$$\mathrm{I}[\{Y_i^{\mathrm{eval}}\}; Y^{\mathrm{acq}} \mid \{x_i^{\mathrm{eval}}\}, x^{\mathrm{acq}}], \tag{JEPIG}$$

$$\mathrm{I}[Y^{\mathrm{eval}}; Y^{\mathrm{acq}} \mid X^{\mathrm{eval}}, x^{\mathrm{acq}}] \tag{EPIG}$$

are examples of such acquisition functions. EPIG and JEPIG are transductive acquisition functions as they depend on $X^{\mathrm{eval}}, \{x_i^{\mathrm{eval}}\}$, respectively.

**Active Sampling.** To increase training efficiency, instead of training with all samples, active sampling (sometimes also called *data pruning*) (Paul et al., 2021) selects the most informative sample $(x^{\mathrm{acq}}, y^{\mathrm{acq}})$ from the training set to train on next. This can be done statically before training the model, in which case this is also referred to as core-set selection, or dynamically, in which case it is also referred to as curriculum learning. The *information gain (IG)*

$$\mathrm{I}[\Omega; y^{\mathrm{acq}} \mid x^{\mathrm{acq}}], \tag{IG}$$

Table 1: *Taxonomy of Information Quantities for Data Subset Selection.* In general, information quantities can be split into ones for active sampling or active learning, into non-transductive and transductive ones, and in the transductive case, into taking an expectation or the joint over (additional) evaluation samples. Here, we show the information quantities for individual acquisition. For batch acquisition, $\{Y_i^{\mathrm{acq}}\}, \{y_i^{\mathrm{acq}}\}, \{x_i^{\mathrm{acq}}\}$ can be substituted.

| | | | Active Learning | | | Active Sampling |
|---|---|---|---|---|---|---|
| Non-Transductive | | EIG/BALD | $\mathrm{I}[\Omega; Y^{\mathrm{acq}} \mid x^{\mathrm{acq}}]$ | | IG | $\mathrm{I}[\Omega; y^{\mathrm{acq}} \mid x^{\mathrm{acq}}]$ |
| Transductive (using $\mathcal{D}^{\mathrm{eval}}$) | Expectation | EPIG | $\mathbb{E}_{\hat{\mathrm{p}}(x^{\mathrm{eval}})} \mathrm{I}[Y^{\mathrm{eval}}; Y^{\mathrm{acq}} \mid x^{\mathrm{eval}}, x^{\mathrm{acq}}]$ | | PIG | $\mathbb{E}_{\hat{\mathrm{p}}(y^{\mathrm{eval}}, x^{\mathrm{eval}})} \mathrm{I}[y^{\mathrm{eval}}; y^{\mathrm{acq}} \mid x^{\mathrm{eval}}, x^{\mathrm{acq}}]$ |
| | Joint | JEPIG | $\mathrm{I}[\{Y_i^{\mathrm{eval}}\}; Y^{\mathrm{acq}} \mid \{x_i^{\mathrm{eval}}\}, x^{\mathrm{acq}}]$ | | JPIG | $\mathrm{I}[\{y_i^{\mathrm{eval}}\}; y^{\mathrm{acq}} \mid \{x_i^{\mathrm{eval}}\}, x^{\mathrm{acq}}]$ |

and *(joint) predictive information gain (JPIG or PIG, respectively)*

$$\mathrm{I}[\{y_i^{\mathrm{eval}}\}; y^{\mathrm{acq}} \mid \{x_i^{\mathrm{eval}}\}, x^{\mathrm{acq}}], \tag{JPIG}$$

$$\mathrm{I}[Y^{\mathrm{eval}}; y^{\mathrm{acq}} \mid X^{\mathrm{eval}}, x^{\mathrm{acq}}] \quad (= \mathbb{E}_{\hat{\mathrm{p}}(x^{\mathrm{eval}}, y^{\mathrm{eval}})} \mathrm{I}[y^{\mathrm{eval}}; y^{\mathrm{acq}} \mid x^{\mathrm{eval}}, x^{\mathrm{acq}}]) \tag{PIG}$$

are examples of such acquisition functions. PIG and JPIG are transductive acquisition functions.

**Batch Acquisition.** In batch active learning, the acquisition function is applied to a batch of candidates $\{x_i^{\mathrm{acq}}\}_{i=1}^B$ and the batch that maximizes the acquisition function is selected for labeling. Similarly, in batch active sampling, the acquisition function is applied to a batch of training samples $\{(x_i^{\mathrm{acq}}, y_i^{\mathrm{acq}})\}_{i=1}^B$ and the batch that maximizes the acquisition function is selected for training. For deep learning, batch acquisition is often the only feasible option in both active learning and active sampling. We extend the above acquisition functions to the batch case by substituting a set of samples in the definitions: $\{x_i^{\mathrm{acq}}\}$ for $x^{\mathrm{acq}}$, $\{Y_i^{\mathrm{acq}}\}$ for $Y^{\mathrm{acq}}$, and $\{y_i^{\mathrm{acq}}\}$ for $y^{\mathrm{acq}}$, respectively, and treating the sets as joint random variables (Kirsch et al., 2019).

**Submodular Acquisition Functions.** Choosing the subset $\{x_i^{\mathrm{acq}}\}$ naively is intractable due to the exponential number of possible acquisition batches. Instead of maximizing the acquisition function on all possible batches, we can often use submodularity (Nemhauser et al., 1978). A set function $f$ is submodular when:

$$f(A \cup B) \leq f(A) + f(B) - f(A \cap B). \tag{submodular}$$

An acquisition batch $\{x_i^{\mathrm{acq}}\}$ can be constructed greedily by selecting the samples that increase the acquisition function the most one-by-one. This greedy algorithm is guaranteed to find a $1 - \frac{1}{e}$-optimal acquisition batch for monotone submodular acquisition functions.

Although the EIG is (monotone) submodular, leading to efficient batch acquisition (Kirsch et al., 2019), the other information quantities (IG, EPIG, JEPIG, PIG) are usually not submodular (Kirsch et al., 2021b). We examine the details of this and compare to the relevant literature in §7.

**Taxonomy of Information Quantities.** Table 1 shows the information quantities along three dimensions: active learning vs active sampling, non-transductive vs transductive, and taking the expectation vs the joint over evaluation samples for transductive information quantities.

**Log Loss.** While many active learning and active sampling methods are motivated independently of the underlying loss, we will focus on log losses, such as the common cross-entropy loss or squared error loss (Gaussian error), as these log losses can be viewed through an information-theoretic or probabilistic lens.

## 3 Second-Order Posterior Approximation

Laplace approximations are a standard tool in Bayesian statistics and machine learning (Daxberger et al., 2021; Immer et al., 2020). In this section, we review the Laplace approximation and introduce it as a special case of a more flexible second-order posterior approximation, a *Gaussian approximation*. It is central to approximating information quantities using observed information, defined in this section, and Fisher information, defined in §4.

Our goal is to approximate the posterior $p(\omega \mid \mathcal{D}, \mathcal{D}^{\text{train}})$ using a (multivariate) Gaussian distribution, where $\mathcal{D} = \{(x_i, y_i)\}_{i=1}^N$ are additional (new) samples, and we start with $p(\omega \mid \mathcal{D}^{\text{train}})$ as the "prior" distribution—we will drop $\mathcal{D}^{\text{train}}$ and use $p(\omega)$ when possible, to shorten the notation.

To begin, we complete the square of a second-order Taylor approximation around the log-parameter likelihood for a fixed $\omega^*$:

$$\log p(\omega) \approx \log p(\omega^*) + \nabla_\omega[\log p(\omega^*)](\omega - \omega^*) + \frac{1}{2}(\omega - \omega^*)^T \nabla_\omega^2[\log p(\omega^*)](\omega - \omega^*) \tag{10}$$

$$= \frac{1}{2}(\omega - (\omega^* - \nabla_\omega^2[\log p(\omega^*)]^{-1}\nabla_\omega[\log p(\omega^*)])^T \nabla_\omega^2[\log p(\omega^*)](\omega - (\omega^* - \nabla_\omega^2[\log p(\omega^*)]^{-1}\nabla_\omega[\log p(\omega^*)]))$$
$$+ \ldots . \tag{11}$$

Importantly, we can express this more concisely by extending the notation of $H[\cdot]$ to its derivatives:

> **Notation 3.1.** *We write* $H'[\cdot]$ *for the Jacobian and* $H''[\cdot]$ *for the Hessian of* $H[\cdot]$:
>
> $$H'[\cdot] \coloneqq -\nabla_\omega \log p(\cdot), \tag{12}$$
> $$H''[\cdot] \coloneqq -\nabla_\omega^2 \log p(\cdot). \tag{13}$$

This notation will be helpful throughout this paper, as both observed information and Fisher information can be expressed in terms of the Hessian of the negative log-parameter likelihood.

Then, we can write:

$$H[\omega] \approx H[\omega^*] + H'[\omega^*](\omega - \omega^*) + \frac{1}{2}(\omega - \omega^*)^T \, H''[\omega^*] \, (\omega - \omega^*) \tag{14}$$

$$= \frac{1}{2}(\omega - (\omega^* - H''[\omega^*]^{-1} \, H'[\omega^*])^T \, H''[\omega^*] \, (\omega - (\omega^* - H''[\omega^*]^{-1} \, H'[\omega^*])) + \ldots . \tag{15}$$

Comparing this to the information content of a multivariate Gaussian distribution:

$$H[\mathcal{N}(w; \mu, \Sigma)] = \frac{1}{2}(\omega - \mu)^T \, \Sigma^{-1} \, (\omega - \mu) + \ldots, \tag{16}$$

we obtain the Gaussian approximation, which we will apply throughout this paper:

**Proposition 3.2.** *The* Gaussian approximation *of the distribution* $p(\omega)$ *of* $\Omega$ *around some* $\omega^*$ *is given by:*

$$\Omega \overset{\approx}{\sim} \mathcal{N}(\omega^* - H''[\omega^*]^{-1} \, H'[\omega^*], \, H''[\omega^*]^{-1}), \tag{17}$$

*where* $H''[\omega^*]$ *must be positive-definite. If* $\omega^*$ *is also a (global) minimizer of* $H[\omega]$ *(that is,* $H'[\omega^*] = 0$*), we obtain the* Laplace approximation:

$$\Omega \overset{\approx}{\sim} \mathcal{N}(\omega^*, \, H''[\omega^*]^{-1}). \tag{18}$$

**Approximation Quality.** However, this approximation can be arbitrarily bad depending on $p(\omega)$ and $\omega^*$. Given enough data, it is often argued that $p(\omega)$ will concentrate around the maximum a posteriori (MAP) estimate, giving rise to the Laplace approximation. In statistics, the Bernstein-von Mises theorem is often used to motivate this, but insufficient data to reach concentration of parameters and multimodality in over-parameterized models (Long, 2021) can be an issue for deep active learning and active sampling.

**Flat Minimum Intuition.** A positive definite Hessian implies that the information content (point-wise entropy) is convex around $\omega^*$ and, equivalently, that the (log) posterior is concave around $\omega^*$. The latter provides an intuition for the Gaussian approximation: the Hessian measures curvature, and the "flatter" the Hessian, e.g., the smaller the largest eigenvalue or the smaller the determinant, the less the loss changes when $\omega^*$ is perturbed. This leads to the search for flat minima as a way to improve generalization (Hinton and Van Camp, 1993; Hochreiter and Schmidhuber, 1994; Smith and Le, 2017).

**Notation 3.3.** *To further shorten the notation, we write* $H''[\mathcal{D} \mid \omega^*]$ *instead of* $H''[\{y_i\} \mid \{x_i\}, \omega^*]$.

**Posterior Approximation of** $\Omega \mid \mathcal{D}$**.** While the Laplace approximation is centered on a (global) minimizer, the Gaussian approximation can be used for a (potentially low-quality) posterior approximation in general. We can expand $\mathrm{H}[\omega^* \mid \mathcal{D}]$ using Bayes' theorem and the additivity of the logarithm. That is, we have:

$$\mathrm{H}[\omega^* \mid \mathcal{D}] = \mathrm{H}[\mathcal{D} \mid \omega^*] + \mathrm{H}[\omega^*] - \mathrm{H}[\mathcal{D}], \tag{19}$$

and then, as $\mathrm{H}[\mathcal{D}]$ is independent of $\omega$:

$$\mathrm{H}'[\omega^* \mid \mathcal{D}] = \mathrm{H}'[\mathcal{D} \mid \omega^*] + \mathrm{H}'[\omega^*] + 0 = \mathrm{H}'[\mathcal{D} \mid \omega^*] + \mathrm{H}'[\omega^*], \tag{20}$$

$$\mathrm{H}''[\omega^* \mid \mathcal{D}] = \mathrm{H}''[\mathcal{D} \mid \omega^*] + \mathrm{H}''[\omega^*]. \tag{21}$$

**Proposition 3.4.** *The* observed information $\mathrm{H}''[\{y_i\} \mid \{x_i\}, \omega^*]$ *is additive:*

$$\mathrm{H}''[\{y_i\} \mid \{x_i\}, \omega^*] = \sum_i \mathrm{H}''[y_i \mid x_i, \omega^*] = \sum_i -\nabla_\omega^2 \log \mathrm{p}(y_i \mid x_i, \omega^*). \tag{22}$$

Note that the observed information has the opposite sign compared to other works because it simplifies the exposition.

**Uninformative Prior.** For a Gaussian prior $\mathrm{p}(\omega) \sim \mathcal{N}(\mu, \Sigma)$, we have $\mathrm{H}''[\omega^*] = \Sigma^{-1}$ and $\mathrm{H}''[\omega^* \mid \mathcal{D}] = \mathrm{H}''[\mathcal{D} \mid \omega^*] + \Sigma^{-1}$. For an uninformative prior with "infinite prior variance" $\Sigma^{-1} \to 0$, we have $\mathrm{H}''[\omega^*] = 0$, and $\mathrm{H}''[\omega^* \mid \mathcal{D}] = \mathrm{H}''[\mathcal{D} \mid \omega^*]$.

> **Proposition 3.5.** *The entropy of the second-order approximation of* $\mathrm{p}(\omega)$ *around* $\omega^*$ *is*
>
> $$\mathrm{H}[\Omega] \approx -\tfrac{1}{2} \log \det \mathrm{H}''[\omega^*] + C_k, \tag{23}$$
>
> *where* $C_k = \frac{k}{2} \log 2\pi e$ *is a constant (independent of* $\mathcal{D}$ *and* $\omega^*$*) and* $k$ *is the number of dimensions of* $\omega$*.*

While Proposition 3.5 is straightforward, it is the main result for this section as it will allow us to approximate all the mentioned information quantities in §5 and §6.

## 4 Fisher Information

Fisher information plays a central role in the approximations of information quantities because, unlike the observed information, it is always positive semidefinite. We use Fisher information to unify various acquisition functions in §7. The following section revisits Fisher information, its properties, special cases, and common approximations. All proofs are given in §A.

In particular, we look at two special cases with more favorable properties: following Kunstner et al. (2019), when we can write our model as $\mathrm{p}(y \mid \hat{z} = \hat{f}(x; \omega))$, where $\hat{f}(x; \omega)$ are the logits, and $\mathrm{p}(y \mid \hat{z})$ is a distribution from the exponential family, Fisher information is independent of $y$, which has useful consequences as we shall see; and following Chaudhuri et al. (2015), when we have a *Generalized Linear Model (GLM)*, observed information also is independent of $y$. The results for the GLM are often applied as an approximation known as *Generalized Gauss-Newton approximation (GGN)*. Together with numerical approximations, such as a diagonal approximation or low-rank factorizations, observed information and Fisher information can then be efficiently approximated for large deep neural networks (Daxberger et al., 2021).

**Definition 4.1.** *The* Fisher information $\mathrm{H}''[Y \mid x, \omega^*]$ *is the expectation over observed information using the model's own predictions* $\mathrm{p}(y \mid x, \omega^*)$ *for a given* $x$ *at* $\omega^*$*:*

$$\mathrm{H}''[Y \mid x, \omega^*] = \mathbb{E}_{\mathrm{p}(y \mid x, \omega^*)}[\mathrm{H}''[y \mid x, \omega^*]]. \tag{24}$$

This notation of the Fisher information is consistent with the notation for information quantities from §2, introduced in Kirsch and Gal (2021), but extended to the observed information: the Fisher is but an expectation over the observed information, and the observed information is the Hessian of the negative log-likelihood.

**Proposition 4.2.** *Like observed information, Fisher information is additive:*

$$\mathrm{H}''[\{Y_i\} \mid \{x_i\}, \omega^*] = \sum_i \mathrm{H}''[\{Y_i\} \mid x_i, \omega^*]. \tag{25}$$

There are two other equivalent definitions of Fisher information:

**Proposition 4.3.** *Fisher information is equivalent to:*

$$\mathrm{H}''[Y \mid x, \omega^*] = \mathbb{E}_{\mathrm{p}(y|x,\omega^*)}[\mathrm{H}'[y \mid x, \omega^*]^T \ \mathrm{H}'[y \mid x, \omega^*]] = \mathrm{Cov}[\mathrm{H}'[Y \mid x, \omega^*]]. \tag{26}$$

**Special Case: Exponential Family.** Kunstner et al. (2019) show in their appendix that if we split a discriminative model into prelogits $\hat{f}(x; \omega)$ and a predictor $\mathrm{p}(y \mid \hat{z} = \hat{f}(x; \omega))$, Fisher information does not depend on $y$ when $\mathrm{p}(y \mid \hat{z})$ is a distribution from an exponential family (independent of $\omega$). This covers a normal distribution for regression parameterized by mean and variance predictions or a categorical distribution via the softmax function. The following statements and proofs follow Kunstner et al. (2019):

**Proposition 4.4.** *The Fisher information* $\mathrm{H}''[Y \mid x, \omega^*]$ *for a model* $\mathrm{p}(y \mid \hat{z} = \hat{f}(x; \omega^*))$ *is equivalent to:*

$$\mathrm{H}''[Y \mid x, \omega^*] = \nabla_\omega \hat{f}(x; \omega^*)^T \ \mathbb{E}_{\mathrm{p}(y|x,\omega^*)}[\nabla_{\hat{z}}^2 \mathrm{H}[y \mid \hat{z} = \hat{f}(x; \omega^*)]] \nabla_\omega \hat{f}(x; \omega^*), \tag{27}$$

*where* $\nabla_{\hat{z}}^2 \mathrm{H}[y \mid \hat{z} = \hat{f}(x; \omega^*)]$ *is short for* $\nabla_{\hat{z}}^2 \mathrm{H}[y \mid \hat{z}]\big|_{\hat{z}=\hat{f}(x;\omega^*)}$.

**Proposition 4.5.** *The Fisher information* $\mathrm{H}''[Y|x, \omega^*]$ *of a model of the form* $\mathrm{p}(y|\hat{z} = \hat{f}(x; \omega^*))$ *is independent of* $y$, *where* $\mathrm{p}(y \mid \hat{z})$ *is a distribution from an exponential family, i.e.,* $\log \mathrm{p}(y \mid \hat{z}) = \hat{z}^T T(y) - A(\hat{z}) + \log h(y)$:

$$\mathrm{H}''[Y \mid x, \omega^*] = \nabla_\omega \hat{f}(x; \omega^*)^T \ \nabla_{\hat{z}}^2 A(\hat{z} = \hat{f}(x; \omega^*)) \ \nabla_\omega \hat{f}(x; \omega^*). \tag{28}$$

*It is crucial that the exponential distribution not depend on* $\omega$. This simplifies computing Fisher information: no expectation over $y$s is needed anymore. The full outer product may not be needed explicitly either.

As examples, we will consider two common parameteric distributions from the exponential family:

**Gaussian Distribution.** When $\mathrm{p}(y \mid \hat{z}) = \mathcal{N}(y; \hat{z}, 1)$, we have $\mathrm{H}''[y \mid \hat{z}] = 1$ for all $y, \hat{z}$, and thus

$$\mathrm{H}''[Y \mid x, \omega^*] = \nabla_\omega \hat{f}(x; \omega^*)^T \ \nabla_\omega \hat{f}(x; \omega^*). \tag{29}$$

**Categorical Distribution.** When $\mathrm{p}(y|\hat{z}) = \mathrm{softmax}(\hat{z})_y$, we have $\mathrm{H}''[y \mid \hat{z}] = \mathrm{diag}(\pi) - \pi \pi^T$, with $\pi_y = \mathrm{p}(y \mid \hat{z})$, and thus:

$$\mathrm{H}''[Y \mid x, \omega^*] = \nabla_\omega \hat{f}(x; \omega^*)^T \ (\mathrm{diag}(\pi) - \pi \pi^T) \ \nabla_\omega \hat{f}(x; \omega^*). \tag{30}$$

**Special Case: Generalized Linear Models.** Chaudhuri et al. (2015) require that observed information is independent of $y$, which we will also use later. This holds for Generalized Linear Models:

**Definition 4.6.** *A generalized linear model (GLM) is a model* $\mathrm{p}(y \mid \hat{z} = \hat{f}(x; \omega))$ *such that* $\log \mathrm{p}(y \mid \hat{z}) = \hat{z}^T T(y) - A(\hat{z}) + \log h(y)$ *is a distribution of the exponential family, independent of* $\omega$, *and* $\hat{f}(x; \omega) = \omega^T x$ *is linear in the parameters* $\omega$.

**Proposition 4.7.** *The observed information* $\mathrm{H}''[y \mid x, \omega^*]$ *of a GLM is independent of* $y$.

$$\mathrm{H}''[y \mid x, \omega^*] = \nabla_\omega \hat{f}(x; \omega^*)^T \nabla_{\hat{z}}^2 \mathrm{H}[y \mid \hat{z} = \hat{f}(x; \omega^*)] \nabla_\omega \hat{f}(x; \omega^*) \tag{31}$$

$$= \nabla_\omega \hat{f}(x; \omega^*)^T \nabla_{\hat{z}}^2 A(w^T x) \nabla_\omega \hat{f}(x; \omega^*). \tag{32}$$

**Proposition 4.8.** *For a model such that the observed information* $\mathrm{H}''[y \mid x, \omega^*]$ *is independent of* $y$, *we have:*

$$\mathrm{H}''[Y \mid x, \omega^*] = \mathrm{H}''[y^* \mid x, \omega^*] \tag{33}$$

*for any* $y^*$, *and also trivially:*

$$\mathrm{H}''[Y \mid x, \omega^*] = \mathbb{E}_{\mathrm{p}(y|x)}[\mathrm{H}''[y \mid x, \omega^*]]. \tag{34}$$

Note that the expectation is over $\mathrm{p}(y \mid x)$ and not $\mathrm{p}(y \mid x, \omega^*)$, and $\mathbb{E}_{\mathrm{p}(\{y_i\}|\{x_i\})}[\mathrm{H}''[\{y_i\} \mid \{x_i\}, \omega^*]] = \mathrm{H}''[\{Y_i\} \mid \{x_i\}, \omega^*]$ is additive then.

**Proposition 4.9.** *For a GLM, when* $\hat{f}(x; \omega) : \mathbb{R}^D \to \mathbb{R}^K$, *where* $K$ *is the number of classes (outputs),* $D$ *is the number of input dimensions,* $\omega \in \mathbb{R}^{D \times K}$, *and assuming the parameters are flattened into a single vector for the Jacobian, we have* $\nabla_\omega \hat{f}(x; \omega) = \mathrm{Id}_K \otimes x^T \in \mathbb{R}^{K \times (K \cdot D)}$, *where* $\otimes$ *denotes the Kronecker product, and:*

$$\nabla_\omega \hat{f}(x; \omega^*)^T \nabla_{\hat{z}}^2 A(\omega^T x) \nabla_\omega \hat{f}(x; \omega^*) = \nabla_{\hat{z}}^2 A(w^T x) \otimes x\, x^T. \tag{35}$$

This property is useful for computing the Fisher information of a GLM in practice (Ash et al., 2021).

$\mathbf{p(y \mid x, \omega^*)}$ **vs** $\mathbf{p(y \mid x)}$**.** Having a GLM solves an important issue we will encounter in §5: approximating the EIG requires taking an expectation over $\mathrm{p}(y \mid x)$ and not $\mathrm{p}(y \mid x, \omega^*)$. One can approximate $\mathrm{p}(y \mid x) \approx \mathrm{p}(y \mid x, \omega^*)$, which can be justified in the limit, but this is probably not a good approximation in the cases interesting for active learning and active sampling. With a GLM, this is not a problem.

**Generalized Gauss-Newton Approximation.** In the case of an exponential family but not a GLM, the equality in Proposition 4.7 is often used as an approximation for the observed information—we simply use the respective Fisher information as an approximation of observed information (via Proposition 4.5):

$$\mathrm{H}''[y \mid x, \omega^*] \approx \mathrm{H}''[Y \mid x, \omega^*] = \nabla_\omega \hat{f}(x; \omega^*)^T \nabla_{\hat{z}}^2 A(w^T x) \nabla_\omega \hat{f}(x; \omega^*). \tag{36}$$

This is known as *Generalized Gauss-Newton (GGN) approximation* (Kunstner et al., 2019; Immer et al., 2020). This approximation has the advantage that it is always positive semidefinite unlike the true Hessian.

**Last-Layer Approaches.** GLMs are often used in deep active learning (Ash et al., 2019; 2021; Kothawade et al., 2022; 2021). If we split the model into $\mathrm{p}(y \mid x, \omega) = \mathrm{p}(y \mid z = \omega^T f(x))$, where $z = f(x)$ are the embeddings and treat the encoder $f(x)$ as fixed, we obtain a GLM based on the weights of the last layer, which uses the embeddings as input.

Armed with this knowledge, we can now derive approximations for the information quantities of interest using observed information and Fisher information and consider their properties. The GGN approximation and last-layer approaches feature heavily in the literature to make computing these approximations more tractable as they reduce computational requirements and memory usage.

# 5 Approximating Information Quantities

We now derive approximations and proxy objectives for information quantities. We base them on observed information and Fisher information introduced in the previous sections. These approximations help us connect the information quantities to existing the literature in non-Bayesian data subset selection in §7.

In particular, we derive approximations for EIG and EPIG as they show the qualitative differences between non-transductive and transductive objectives, and compare the approximations of the IG and EIG: importantly, there is no difference between the latter when we use a GLM or the GGN approximation. This covers two of the three dimensions in Table 1. We examine JEPIG and the other quantities in the appendix in §B. We find that the trace approximations of the EPIG and JEPIG objective matches, suggesting that using the trace approximations might be too loose an approximation to capture important qualities of EPIG (Kirsch et al., 2021b). Additional derivations and details can also be found in §B. All this leads to Figure 1, which relates the different approximations to each other and shows that they follow simple patterns.

## 5.1 Approximate Expected Information Gain

The expected information gain is a popular acquisition function in Bayesian optimal experimental design (Lindley, 1956) and in active learning, where it is also known as BALD (Houlsby et al., 2011; Gal et al., 2017).

We can approximate the EIG $I[\Omega; \{Y_i^{\mathrm{acq}}\} \mid \{x_i^{\mathrm{acq}}\}]$ of acquisition candidates $\{x_i^{\mathrm{acq}}\}$ using Gaussian approximations:

$$I[\Omega; \{Y_i^{\mathrm{acq}}\} \mid \{x_i^{\mathrm{acq}}\}] = H[\Omega] - H[\Omega \mid \{Y_i^{\mathrm{acq}}\}, \{x_i^{\mathrm{acq}}\}] \tag{37}$$

$$= H[\Omega] - \mathbb{E}_{\mathrm{p}(\{y_i^{\mathrm{acq}}\}|\{x_i^{\mathrm{acq}}\})}[H[\Omega \mid \{y_i^{\mathrm{acq}}\}, \{x_i^{\mathrm{acq}}\}]] \tag{38}$$

$$\approx -\tfrac{1}{2}\log\det H''[\omega^*] - \mathbb{E}_{\mathrm{p}(\{y_i^{\mathrm{acq}}\}|\{x_i^{\mathrm{acq}}\})}[-\tfrac{1}{2}\log\det H''[\omega \mid \{y_i^{\mathrm{acq}}\}, \{x_i^{\mathrm{acq}}\}]] \tag{39}$$

$$= \tfrac{1}{2}\mathbb{E}_{\mathrm{p}(\{y_i^{\mathrm{acq}}\}|\{x_i^{\mathrm{acq}}\})}[\log\det\left((H''[\{y_i^{\mathrm{acq}}\} \mid \{x_i^{\mathrm{acq}}\}, \omega^*] + H''[\omega^*])\, H''[\omega^*]^{-1}\right)] \tag{40}$$

$$= \tfrac{1}{2}\mathbb{E}_{\mathrm{p}(\{y_i^{\mathrm{acq}}\}|\{x_i^{\mathrm{acq}}\})}[\log\det\left(H''[\{y_i^{\mathrm{acq}}\} \mid \{x_i^{\mathrm{acq}}\}, \omega^*]\, H''[\omega^*]^{-1} + Id\right)]. \tag{41}$$

using Proposition 3.5 twice, where the constant $C_k$ cancels out in eq. (39) as we subtract two entropy terms.

**Generalized Linear Model.** When we have a GLM, we can use Proposition 4.8 to obtain:

$$I[\Omega; \{Y_i^{\mathrm{acq}}\} \mid \{x_i^{\mathrm{acq}}\}] \approx \ldots = \tfrac{1}{2}\mathbb{E}_{\mathrm{p}(\{y_i^{\mathrm{acq}}\}|\{x_i^{\mathrm{acq}}\})}[\log\det\left(H''[\{y_i^{\mathrm{acq}}\} \mid \{x_i^{\mathrm{acq}}\}, \omega^*]\, H''[\omega^*]^{-1} + Id\right)] \tag{42}$$

$$= \tfrac{1}{2}\log\det\left(H''[\{Y_i^{\mathrm{acq}}\} \mid \{x_i^{\mathrm{acq}}\}, \omega^*]\, H''[\omega^*]^{-1} + Id\right). \tag{43}$$

We can upper-bound the log determinant and obtain:

$$\leq \tfrac{1}{2}\mathrm{tr}\left(H''[\{Y_i^{\mathrm{acq}}\} \mid \{x_i^{\mathrm{acq}}\}, \omega^*]\, H''[\omega^*]^{-1}\right) \tag{44}$$

$$= \tfrac{1}{2}\sum_i \mathrm{tr}\left(H''[\{Y_i^{\mathrm{acq}}\} \mid x^{\mathrm{acq}}{}_i, \omega^*]\, H''[\omega^*]^{-1}\right). \tag{45}$$

where we have used the following inequality (proof in §B.1):

**Lemma 5.1.** *For symmetric, positive semidefinite matrices $A$, we have (with equality iff $A = 0$):*

$$\log\det(A + Id) \leq \mathrm{tr}(A). \tag{46}$$

**General Case & Exponential Family.** For the general case, we need to make a strong approximation:

$$\mathrm{p}(\{y_i^{\mathrm{acq}}\} \mid \{x_i^{\mathrm{acq}}\}) \approx \mathrm{p}(\{y_i^{\mathrm{acq}}\} \mid \{x_i^{\mathrm{acq}}\}, \omega^*), \tag{47}$$

which might hold for a mostly converged posterior but probably not in cases with little data. This turns the approximation into an upper bound. Alternatively, we could use the GGN approximation when we have an exponential family for the same result (but not an upper bound). See §B.1 for the derivation.

**Proposition 5.2** (EIG)**.** *The expected information gain can be approximately upper bounded via:*

$$\mathrm{I}[\Omega; \{Y^{acq}_i\} \mid \{x^{acq}_i\}, \mathcal{D}^{train}] \overset{\approx}{\leq} \tfrac{1}{2}\log\det\left(\sum_i \mathrm{H}''[Y^{acq}_i \mid x^{acq}_i, \omega^*]\,\mathrm{H}''[\omega^* \mid \mathcal{D}^{train}]^{-1} + Id\right) \quad (48)$$

$$\leq \tfrac{1}{2}\sum_i \mathrm{tr}\left(\mathrm{H}''[Y^{acq}_i \mid x^{acq}_i, \omega^*]\,\mathrm{H}''[\omega^* \mid \mathcal{D}^{train}]^{-1}\right). \quad (49)$$

*Furthermore, we have the following proxy objective:*

$$\underset{\{x^{acq}_i\}}{\arg\max}\,\mathrm{I}[\Omega; \{Y^{acq}_i\} \mid \{x^{acq}_i\}, \mathcal{D}^{train}] = \underset{\{x^{acq}_i\}}{\arg\max}\,-\mathrm{H}[\Omega \mid \{Y^{acq}_i\}, \{x^{acq}_i\}, \mathcal{D}^{train}]\ , \ and \quad (50)$$

$$-\mathrm{H}[\Omega \mid \{Y^{acq}_i\}, \{x^{acq}_i\}, \mathcal{D}^{train}] \overset{\approx}{\leq} \tfrac{1}{2}\log\det\left(\sum_i \mathrm{H}''[Y^{acq}_i \mid x^{acq}_i, \omega^*] + \mathrm{H}''[\omega^* \mid \mathcal{D}^{train}]\right) - C_k. \quad (51)$$

The second statement follows from Equation (37), since $\mathrm{H}[\Omega \mid \mathcal{D}^{\mathrm{train}}]$ is constant and provides a proxy objective when we are only interested in optimizing the EIG. In §7, we connect it to the expected gradient length approach in active learning and show that an ablation in Ash et al. (2021) examines the wrong objective.

**Batch Acquisition Pathologies.** Importantly, this approximation of the EIG using the trace is additive, whereas the one using the log determinant is not. This means that the trace approximation ignores the dependencies between the samples and can only lead to naive top-k batch acquisition; see Kirsch et al. (2019; 2021a) for details of the pathologies of top-k batch acquisition.

## 5.2 Approximate Information Gain

Following the same steps, we can also approximate the information gain, which is useful for active sampling:

**Proposition 5.3** (IG)**.** *The information gain* $\mathrm{I}[\Omega; \{y^{acq}_i\} \mid \{x^{acq}_i\}, \mathcal{D}^{train}] = \mathrm{H}[\Omega \mid \mathcal{D}^{train}] - \mathrm{H}[\Omega \mid \{y^{acq}_i\}, \{x^{acq}_i\}, \mathcal{D}^{train}]$ *can be approximately upper bounded via:*

$$\mathrm{I}[\Omega; \{y^{acq}_i\} \mid \{x^{acq}_i\}, \mathcal{D}^{train}] \approx \tfrac{1}{2}\log\det\left(\mathrm{H}''[\{y^{acq}_i\} \mid \{x^{acq}_i\}, \omega^*]\,\mathrm{H}''[\omega^* \mid \mathcal{D}^{train}]^{-1} + Id\right) \quad (52)$$

$$\leq \tfrac{1}{2}\sum_i \mathrm{tr}\left(\mathrm{H}''[y^{acq}_i \mid x^{acq}_i, \omega^*]\,\mathrm{H}''[\omega^* \mid \mathcal{D}^{train}]^{-1}\right). \quad (53)$$

*Furthermore, we have the following proxy objective:*

$$\underset{\{x^{acq}_i\}}{\arg\max}\,\mathrm{I}[\Omega; \{y^{acq}_i\} \mid \{x^{acq}_i\}, \mathcal{D}^{train}] = \underset{\{x^{acq}_i\}}{\arg\max}\,-\mathrm{H}[\Omega \mid \{y^{acq}_i\}, \{x^{acq}_i\}, \mathcal{D}^{train}], \ and \quad (54)$$

$$-\mathrm{H}[\Omega \mid \{y^{acq}_i\}, \{x^{acq}_i\}, \mathcal{D}^{train}] \approx \tfrac{1}{2}\log\det\left(\mathrm{H}''[\{y^{acq}_i\} \mid \{x^{acq}_i\}, \omega^*] + \mathrm{H}''[\omega^* \mid \mathcal{D}^{train}]\right) - C_k. \quad (55)$$

**Comparison to EIG.** Importantly, when we have a GLM or use the GGN approximation, this approximation of the IG is equal to the one of the EIG. This tells us that active learning on a GLM with the EIG approximation will work as well as if we had access to the labels. Equivalently, active sampling via IG with the GGN approximation will not work better than the respective active learning approach.

## 5.3 Approximate (Joint) Expected Predictive Information Gain

In transductive active learning, we have access to an (empirical) distribution $\hat{\mathrm{p}}(x^{\mathrm{eval}})$, e.g., the pool set, and want to find the $\{x^{\mathrm{acq}}_i\}$ that maximize the *expected predictive information gain (EPIG)* (Kirsch et al., 2021b). The approximations here will help us connect BAIT (Ash et al., 2021) to EPIG. For simplicity, we consider the non-batch case here. The batch case can be handled analogously. The EPIG objective is defined as:

$$\underset{x^{\mathrm{acq}}}{\arg\max}\,\mathrm{I}[Y^{\mathrm{eval}}; Y^{\mathrm{acq}} \mid X^{\mathrm{eval}}, x^{\mathrm{acq}}] = \underset{x^{\mathrm{acq}}}{\arg\max}\,\mathbb{E}_{\hat{\mathrm{p}}(x^{\mathrm{eval}})}\,\mathrm{I}[Y^{\mathrm{eval}}; Y^{\mathrm{acq}} \mid x^{\mathrm{eval}}, x^{\mathrm{acq}}], \quad (56)$$

We expand the objective as follows:

$$\mathrm{I}[Y^{\text{eval}}; Y^{\text{acq}} \mid X^{\text{eval}}, x^{\text{acq}}] = \mathrm{I}[\Omega; Y^{\text{eval}} \mid X^{\text{eval}}] - \mathrm{I}[\Omega; Y^{\text{eval}} \mid X^{\text{eval}}, Y^{\text{acq}}, x^{\text{acq}}], \tag{57}$$

where $\mathrm{I}[\Omega; Y^{\text{eval}} \mid X^{\text{eval}}]$ can be removed from the objective because it is independent of $x^{\text{acq}}$. Thus, optimizing EPIG is equivalent to *minimizing* $\mathrm{I}[\Omega; Y^{\text{eval}} \mid X^{\text{eval}}, Y^{\text{acq}}, x^{\text{acq}}]$:

$$\underset{x^{\text{acq}}}{\arg\max}\, \mathrm{I}[Y^{\text{eval}}; Y^{\text{acq}} \mid X^{\text{eval}}, x^{\text{acq}}] = \underset{x^{\text{acq}}}{\arg\min}\, \mathrm{I}[\Omega; Y^{\text{eval}} \mid X^{\text{eval}}, Y^{\text{acq}}, x^{\text{acq}}]. \tag{58}$$

Following Proposition 5.2, this can be approximated by:

$$\mathrm{I}[\Omega; Y^{\text{eval}} \mid X^{\text{eval}}, Y^{\text{acq}}, x^{\text{acq}}]$$
$$\approx \tfrac{1}{2}\, \mathbb{E}_{\mathrm{p}(y^{\text{eval}}, y^{\text{acq}} \mid x^{\text{eval}}, x^{\text{acq}})\, \hat{\mathrm{p}}(x^{\text{eval}})}[\log\det\left(\mathrm{H}''[y^{\text{eval}} \mid x^{\text{eval}}, \omega^*]\,(\mathrm{H}''[y^{\text{acq}} \mid x^{\text{acq}}, \omega^*] + \mathrm{H}''[\omega^*])^{-1} + Id)]. \tag{59}$$

**Generalized Linear Model.** For a generalized linear model, we can drop the expectation and obtain:

$$\mathrm{I}[\Omega; Y^{\text{eval}} \mid X^{\text{eval}}, Y^{\text{acq}}, x^{\text{acq}}]$$
$$\approx \tfrac{1}{2}\, \mathbb{E}_{\hat{\mathrm{p}}(x^{\text{eval}})}[\log\det\left(\mathrm{H}''[Y^{\text{eval}} \mid x^{\text{eval}}, \omega^*]\,(\mathrm{H}''[Y^{\text{acq}} \mid x^{\text{acq}}, \omega^*] + \mathrm{H}''[\omega^*])^{-1} + Id)] \tag{60}$$
$$\leq \tfrac{1}{2}\log\det\left(\mathbb{E}_{\hat{\mathrm{p}}(x^{\text{eval}})}[\mathrm{H}''[Y^{\text{eval}} \mid x^{\text{eval}}, \omega^*]]\,(\mathrm{H}''[Y^{\text{eval}} \mid x^{\text{acq}}, \omega^*] + \mathrm{H}''[\omega^*])^{-1} + Id\right) \tag{61}$$
$$\leq \tfrac{1}{2}\mathrm{tr}\left(\mathbb{E}_{\hat{\mathrm{p}}(x^{\text{eval}})}[\mathrm{H}''[Y^{\text{eval}} \mid x^{\text{eval}}, \omega^*]]\,(\mathrm{H}''[Y^{\text{acq}} \mid x^{\text{acq}}, \omega^*] + \mathrm{H}''[\omega^*])^{-1}\right), \tag{62}$$

where we have used the concavity of the log determinant and Lemma 5.1.

**General Case & Exponential Family.** To our knowledge, there is no rigorous way to obtain a similar result in the general case as the Fisher information for an acquisition candidate now lies within an inverted term. Of course, the GGN approximation can be applied when we have an exponential family, which leads to the GLM result above as an approximation. See §B.2 for more details.

---

**Proposition 5.4** (EPIG). *For a generalized linear model (or with the GGN approximation), we have:*

$$\underset{\{x_i^{acq}\}}{\arg\max}\, \mathrm{I}[Y^{eval}; \{Y_i^{acq}\} \mid X^{eval}, \{x_i^{acq}\}, \mathcal{D}^{train}] = \underset{\{x_i^{acq}\}}{\arg\min}\, \mathrm{I}[\Omega; Y^{eval} \mid X^{eval}, \{Y_i^{acq}\}, \{x_i^{acq}\}, \mathcal{D}^{train}], \tag{63}$$

*with*

$$\mathrm{I}[\Omega; Y^{eval} \mid X^{eval}, \{Y_i^{acq}\}, \{x_i^{acq}\}, \mathcal{D}^{train}]$$
$$\approx \mathbb{E}_{\hat{\mathrm{p}}(x^{eval})}[\log\det\left(\mathrm{H}''[Y^{eval} \mid x^{eval}, \omega^*]\,(\mathrm{H}''[\{Y_i^{acq}\} \mid \{x_i^{acq}\}, \omega^*] + \mathrm{H}''[\omega^*])^{-1} + Id)] \tag{64}$$
$$\leq \tfrac{1}{2}\log\det\left(\mathbb{E}_{\hat{\mathrm{p}}(x^{eval})}[\mathrm{H}''[Y^{eval} \mid x^{eval}, \omega^*]]\,(\mathrm{H}''[\{Y_i^{acq}\} \mid \{x_i^{acq}\}, \omega^*] + \mathrm{H}''[\omega^* \mid \mathcal{D}^{train}])^{-1} + Id\right) \tag{65}$$
$$\leq \tfrac{1}{2}\mathrm{tr}\left(\mathbb{E}_{\hat{\mathrm{p}}(x^{eval})}[\mathrm{H}''[Y^{eval} \mid x^{eval}, \omega^*]]\,(\mathrm{H}''[\{Y_i^{acq}\} \mid \{x_i^{acq}\}, \omega^*] + \mathrm{H}''[\omega^* \mid \mathcal{D}^{train}])^{-1}\right). \tag{66}$$

---

**Batch Acquisition Pathologies.** Unlike for the EIG, the trace approximation of EPIG is not additive in $\{x_i^{\text{acq}}\}$, and we cannot conclude that it suffers from batch acquisition pathologies like the trace approximation of the EIG.

**Approximations for JEPIG, PIG and JPIG.** We can follow the same derivation for JEPIG:

$$\mathrm{I}[\Omega; \{Y_i^{\text{eval}}\} \mid \{x_i^{\text{eval}}\}, Y^{\text{acq}}, x^{\text{acq}}] \tag{67}$$
$$\approx \tfrac{1}{2}\log\det\left(\mathbb{E}_{\mathrm{p}(\{y_i^{\text{eval}}\}, y^{\text{acq}} \mid \{x_i^{\text{eval}}\}, x^{\text{acq}})}[\mathrm{H}''[\{y_i^{\text{eval}}\} \mid \{x_i^{\text{eval}}\}, \omega^*]]\,(\mathrm{H}''[y^{\text{acq}} \mid x^{\text{acq}}, \omega^*] + \mathrm{H}''[\omega^*])^{-1} + Id\right).$$

Then, applying the steps after eq. (61), we can devise similar approximations. PIG and JPIG follow the same pattern. Details can be found in §B.3 and §B.4. But how do all these approximations relate to each other?

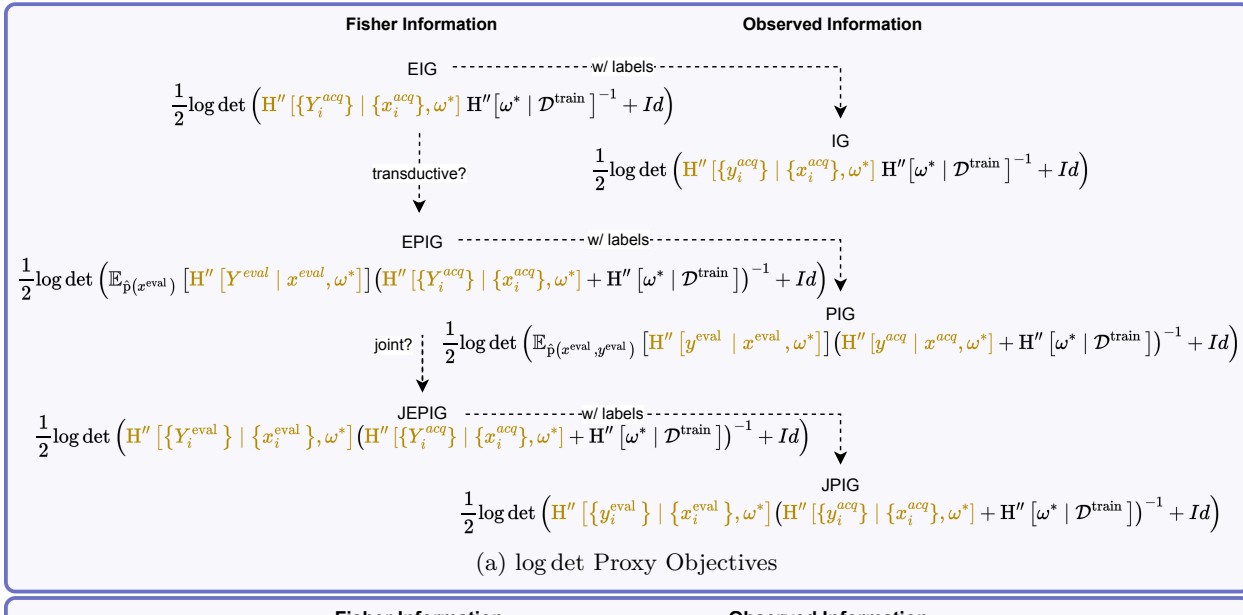

(a) log det Proxy Objectives

(b) Matrix Trace Proxy Objectives

Figure 1: *Comparison of the Approximations/Proxy Objectives.* The difference between active learning and active sampling objectives is in using the Fisher information, which is label independent, or the observed information, which uses label information. JEPIG and EPIG have equivalent proxy objectives when using the matrix trace; see §B.4. The notation makes it obvious that in the GLM case, or when the GGN approximation is used, active learning and active sampling approximations match as $H''[y \mid x, \omega^*] = (\text{resp.} \approx) H''[Y \mid x, \omega^*]$.

## 5.4 Comparison of the Different Information Quantity Approximations

Figure 1 compares the different information quantity approximations for both the log-determinant and trace approximations. We empirically compare the approximations with prediction-space methods in §E. Importantly, the trace approximations of (E)PIG match those of J(E)PIG up to a constant factor (unlike the log-determinant approximations); see §B.4 for details.

Kirsch et al. (2021b) argue that JEPIG converges to BALD in the data limit of the evaluation set—when there are no outliers in the pool set— while EPIG does not. The trace approximation is too strong to preserve this difference. Does this difference matter in practice? We leave this for future work.

Crucially, for GLMs or when using the GGN approximation, the respective active learning and active sampling objectives (EIG and IG, etc.) are equivalent as Fisher information and observed information are the same. In contrast, in the general case, the approximations for EPIG and JEPIG do not have a principled derivation.

## 6 Similarity Matrices and One-Sample Approximations of the Fisher Information

Many data subset selection methods (Iyer et al., 2021; Kothawade et al., 2022; 2021; Ash et al., 2019) use similarity matrices of the loss Jacobians $\mathrm{H}'[\hat{y} \mid x]$, where $\hat{y}$ is usually a hypothesized pseudo-label: often the arg max prediction of the model for $x$. Here, we connect such similarity matrices to the Fisher information and the approximations of information quantities from §5. The proofs are given in §C. Together with §5, this section provides a unified framework for understanding the approximations of information quantities using Fisher information and lays the foundation for the next section, which will connect the cited works in §1 to the approximations of information quantities.

**Connection to Fisher Information.** Crucially, given $\mathcal{D} = \{(y_i, x_i)_i\}$, if we let

$$\hat{\mathrm{H}}'[\mathcal{D} \mid \omega^*] := \begin{pmatrix} \vdots \\ \mathrm{H}'[y_i \mid x_i, \omega^*] \\ \vdots \end{pmatrix} \tag{68}$$

be a "data matrix" of the Jacobians, then $\hat{\mathrm{H}}'[\mathcal{D} \mid \omega^*]\hat{\mathrm{H}}'[\mathcal{D} \mid \omega^*]^T$ gives the similarity matrix $S[\mathcal{D} \mid \omega^*]$ using the Euclidean inner product:

$$S[\mathcal{D} \mid \omega]_{ij} := \langle \mathrm{H}'[y_i \mid x_i, \omega^*], \mathrm{H}'[y_j \mid x_j, \omega^*] \rangle = \hat{\mathrm{H}}'[\mathcal{D} \mid \omega^*]\hat{\mathrm{H}}'[\mathcal{D} \mid \omega^*]^T. \tag{69}$$

Sampling $\{y_i\} \sim \mathrm{p}(\{y_i\} \mid \{x_i\}, \omega^*)$, the "flipped" product $\hat{\mathrm{H}}'[\mathcal{D} \mid \omega^*]^T \hat{\mathrm{H}}'[\mathcal{D} \mid \omega^*]$ yields a *one-sample estimate* of the Fisher information $\mathrm{H}''[\{Y_i\} \mid \{x_i\}, \omega^*]$:

$$\mathrm{H}''[\{Y_i\} \mid \{x_i\}, \omega^*] = \sum_i \mathrm{H}''[Y_i \mid x_i, \omega^*] = \mathbb{E}_{\mathrm{p}(\{y_i\}\mid\{x_i\},\omega^*)} \sum_i \mathrm{H}'[y_i \mid x_i, \omega^*]^T \, \mathrm{H}'[y_i \mid x_i, \omega^*] \tag{70}$$

$$= \mathbb{E}_{\mathrm{p}(\{y_i\}\mid\{x_i\},\omega^*)} \hat{\mathrm{H}}'[\mathcal{D} \mid \omega^*]^T \hat{\mathrm{H}}'[\mathcal{D} \mid \omega^*]. \tag{71}$$

**Hard Pseudo-Labels.** Importantly, using the arg max class for $y_i$, we only obtain a biased estimate (Kunstner et al., 2019, §B).

**Connection to the Expected Information Gain.** When we define an inner product $\langle \cdot, \cdot \rangle_{\mathrm{H}''[\omega^* \mid \mathcal{D}^{\mathrm{train}}]}$ using the Hessian, we can connect the similarity matrix, which uses this inner product:

$$S_{\mathrm{H}''[\omega^*\mid\mathcal{D}^{\mathrm{train}}]}[\mathcal{D} \mid \omega^*] := \hat{\mathrm{H}}'[\mathcal{D} \mid \omega^*] \, \mathrm{H}''[\omega^* \mid \mathcal{D}^{\mathrm{train}}]^{-1} \hat{\mathrm{H}}'[\mathcal{D} \mid \omega^*]^T, \tag{72}$$

to our information gain approximations.

Specifically, we apply the matrix-determinant lemma $\det(AB + M) = \det M \det(Id + BM^{-1}A)$ to obtain:

> **Proposition 6.1.** *Given $\mathcal{D}^{train}$, $\{x_i^{acq}\}$ and (sampled) $\{y_i^{acq}\}$, we have for the EIG:*
>
> $$\mathrm{I}[\Omega; \{Y_i^{acq}\} \mid \{x_i^{acq}\}, \mathcal{D}^{train}] \overset{\approx}{\leq} \tfrac{1}{2}\log\det\left(S_{\mathrm{H}''[\omega^*\mid\mathcal{D}^{train}]}[\mathcal{D}^{acq} \mid \omega^*] + Id\right) \tag{73}$$
> $$\leq \tfrac{1}{2}\mathrm{tr}\, S_{\mathrm{H}''[\omega^*\mid\mathcal{D}^{train}]}[\mathcal{D}^{acq} \mid \omega^*] \tag{74}$$
>
> **Proposition 6.2.** *Assuming an uninformative posterior $\mathrm{H}''[\omega^* \mid \mathcal{D}^{train}] = \lambda Id$ for $\lambda \to 0$, and given $\mathcal{D}^{train}$, $\{x_i^{acq}\}$, and (sampled) $\{y_i^{acq}\}$, we have for the EIG (before taking $\lambda \to 0$):*
>
> $$\mathrm{I}[\Omega; \{Y_i^{acq}\} \mid \{x_i^{acq}\}, \mathcal{D}^{train}] \overset{\approx}{\leq} \tfrac{1}{2}\log\det\left(S[\mathcal{D}^{acq} \mid \omega^*] + \lambda Id\right) - \tfrac{|\mathcal{D}^{acq}|}{2}\log\lambda. \tag{75}$$
>
> *As the second term is independent of $\{x_i^{acq}\}$, we can use the following proxy objective in the limit:*
>
> $$\frac{1}{2}\log\det\left(S[\mathcal{D}^{acq} \mid \omega^*]\right). \tag{76}$$

**Connection to Other Approximate Information Quantities.** Interestingly, we can use the above to obtain approximations of the predictive information gains (EPIG and JEPIG) because the terms that would tend towards $-\infty$ cancel out; see §C for details. For EPIG, we have:

> **Proposition 6.3.** *Given $\mathcal{D}^{eval}, \mathcal{D}^{train}, \{x_i^{acq}\}$ and (sampled) $\{y_i^{acq}\}$, we have for the EPIG:*
>
> $\mathrm{I}[Y^{eval}; \{Y_i^{acq}\} \mid X^{eval}, \{x_i^{acq}\}, \mathcal{D}^{train}]$
>
> $\approx \frac{1}{2}\log\det\left(S_{\mathrm{H}''[\omega^*\mid\mathcal{D}^{train}]}[\mathcal{D}^{eval} \mid \omega^*] + Id\right) - \frac{1}{2}\log\det\left(S_{\mathrm{H}''[\omega^*\mid\mathcal{D}^{train}]}[\mathcal{D}^{acq}, \mathcal{D}^{eval} \mid \omega^*] + Id\right) \qquad (77)$
>
> $\qquad + \frac{1}{2}\log\det\left(S_{\mathrm{H}''[\omega^*\mid\mathcal{D}^{train}]}[\mathcal{D}^{acq} \mid \omega^*] + Id\right). \qquad (78)$
>
> *For an uninformative prior, we have:*
>
> $\approx \frac{1}{2}\log\det\left(S[\mathcal{D}^{eval} \mid \omega^*]\right) - \frac{1}{2}\log\det\left(S[\mathcal{D}^{acq}, \mathcal{D}^{eval} \mid \omega^*]\right) + \frac{1}{2}\log\det\left(S[\mathcal{D}^{acq} \mid \omega^*]\right). \qquad (79)$
>
> *We can drop the terms that only depend on $\mathcal{D}^{eval}$ when we are interested in proxy objectives for optimization.*

These results help connect the objectives of PRISM and SIMILAR to the EIG and EPIG in the next section.

# 7 Connection to Other Acquisition Functions in the Literature

Now, we can connect approaches in non-Bayesian literature to information quantities. Additional proofs are given in §D.

## 7.1 BAIT in "Gone Fishing" (Ash et al., 2021), ActiveSetSelect in "Convergence Rates of Active Learning for Maximum Likelihood Estimation" (Chaudhuri et al., 2015), and EPIG (Kirsch et al., 2021b)

Ash et al. (2021) introduce the *BAIT* objective for deep active learning:

$$\underset{\{x_i^{\mathrm{acq}}\}}{\arg\min}\, \mathrm{tr}\left(\left(\mathrm{H}''[\{Y_i^{\mathrm{acq}}\} \mid \{x_i^{\mathrm{acq}}\}, \omega^*] + \mathrm{H}''[Y^{\mathrm{train}} \mid x^{\mathrm{train}}, \omega^*] + \lambda I\right)^{-1} \mathrm{H}''[\{Y_i^{\mathrm{eval}}\} \mid \{x_i^{\mathrm{eval}}\}, \omega^*]\right), \quad \text{(BAIT)}$$

where $\lambda$ is a hyperparameter[3].

BAIT is based on a similar objective for GLMs from Chaudhuri et al. (2015). While Ash et al. (2021) apply this objective to DNNs, they only use the last layer to approximate the Fisher information. The last layer, with appropriate activation functions and losses, constitutes a GLM as seen in §4.

Following Proposition 5.4, we immediately see that Ash et al. (2021) perform transductive active learning (using the pool set as an evaluation set) and approximate a proxy objective for (J)EPIG:

> **Proposition 7.1.** *Both Ash et al. (2021) and Chaudhuri et al. (2015) perform transductive active learning, approximating (J)EPIG (Kirsch et al., 2021b; MacKay, 1992) using a last-layer approach (or GLM):*
>
> $\underset{x^{acq}}{\arg\max}\, \mathrm{I}[Y^{eval}; \{Y_i^{acq}\} \mid X^{eval}, \{x_i^{acq}\}]$
>
> $\approx \underset{x^{acq}}{\arg\min}\, \mathrm{tr}(\mathrm{H}''[\{Y_i^{eval}\} \mid \{x_i^{eval}\}, \omega^*]\, (\mathrm{H}''[\{Y_i^{acq}\} \mid \{x_i^{acq}\}, \omega^*] + \mathrm{H}''[\omega^* \mid \mathcal{D}^{train}])^{-1}), \qquad (80)$
>
> *with $\mathrm{H}''[\omega^* \mid \mathcal{D}^{train}] = \mathrm{H}''[\mathcal{D}^{train} \mid \omega^*] + \mathrm{H}''[\omega^*]$ and $\mathrm{H}''[\omega^*] = \lambda\, Id$.*

*Proof.* This follows immediately for GLM (last-layer approaches) when we expand $\mathrm{H}''[\omega^* \mid \mathcal{D}^{\mathrm{train}}]$. Chaudhuri et al. (2015) in particular uses an uninformative prior, that is $\lambda = 0$. Comparing the resulting objectives yields the statement. $\square$

---

[3]This is the BAIT objective as computed in Algorithm 1 in Ash et al. (2021) and in the published implementation https://github.com/JordanAsh/badge/blob/master/query_strategies/bait_sampling.py.

Thus, Ash et al. (2021) and Kirsch et al. (2021b) employ the same underlying acquisition function, albeit using very different approaches: Ash et al. (2021) use a last-layer Fisher information matrix, whereas Kirsch et al. (2021b) use approximate BNNs and sample joint predictions.

**Research Questions.** EPIG is not submodular, and the greedy selection of an acquisition batch does not come with any optimality guarantees. While Kirsch et al. (2021b) ignore this, Ash et al. (2021) propose a heuristic that empirically performs better: They greedily select additional acquisition candidates in forward pass (twice the intended batch acquisition size) and then greedily remove the least informative samples from the batch in a backward pass. *Would this heuristic also prove beneficial for all the other information quantities that are not submodular?*

While Ash et al. (2021) state that they only use the last-layer approach for performance reasons, following §4, it does not seem that this approach translates beyond a last-layer approach for DNNs in a principled fashion (see §B.2). *Is there a principled approach for the general case that goes beyond last-layer active learning when using Fisher information without the GGN approximation?*

Ash et al. (2021) ablate trace and determinantal approaches, similar to comparing eq. (66) and eq. (65), yet they do not include $+Id$ in the log determinant expression, which leads them to examine the EIG in their ablation[4]:

$$\underset{x^{\mathrm{acq}}}{\arg\min} \log\det(\mathrm{H}''[\{Y_i^{\mathrm{eval}}\} \mid \{x_i^{\mathrm{eval}}\}, \omega^*] \, (\sum_i \mathrm{H}''[Y^{\mathrm{acq}}_i \mid x^{\mathrm{acq}}_i, \omega^*] + \mathrm{H}''[\omega^* \mid \mathcal{D}^{\mathrm{train}}])^{-1})$$

$$= \underset{x^{\mathrm{acq}}}{\arg\min} \log\det \mathrm{H}''[\{Y_i^{\mathrm{eval}}\} \mid \{x_i^{\mathrm{eval}}\}, \omega^*] - \log\det(\sum_i \mathrm{H}''[Y^{\mathrm{acq}}_i \mid x^{\mathrm{acq}}_i, \omega^*] + \mathrm{H}''[\omega^* \mid \mathcal{D}^{\mathrm{train}}]) \quad (81)$$

$$= \underset{x^{\mathrm{acq}}}{\arg\max} \log\det(\sum_i \mathrm{H}''[Y^{\mathrm{acq}}_i \mid x^{\mathrm{acq}}_i, \omega^*] + \mathrm{H}''[\omega^* \mid \mathcal{D}^{\mathrm{train}}]) + C, \quad (82)$$

and the last term matches the EIG in eq. (65) up constant terms independent of $x^{\mathrm{acq}}$. Thus, the ablations in Ash et al. (2021) only compares EIG and EPIG. *Could comparing eq. (66) and eq. (65) provide more insightful results about the trade-offs between trace and determinant approximations?*

### 7.2  BADGE (Ash et al., 2019) and BatchBALD (Kirsch et al., 2019)

BADGE performs batch acquisition using a similarity matrix: Using the concepts of §6, BADGE uses hard pseudo-labels together with last-layer gradient embeddings for the similarity matrix $S[\mathcal{D}^{\mathrm{acq}} \mid \omega^*]$. The authors sample from a k-DPP (Kulesza and Taskar, 2011) based on this similarity matrix to select a diverse batch of samples for acquisition. However, to further speed up acquisitions, BADGE uses k-MEANS++ (Arthur and Vassilvitskii, 2006; Ostrovsky et al., 2013) instead of a k-DPP: it uses the Jacobians $\mathrm{H}'[y \mid x, \omega^*]$ of the data matrix directly and samples a diverse batch based on the Euclidean distance between these Jacobians. However, sampling from k-DPPs does not pick the most informative batch overall, and the ablations in Ash et al. (2019) show that k-MEANS++ outperforms k-DPP. Finally, the paper only motivates using gradient embeddings with hard pseudo-labels through intuitions: the gradient length captures information about the model's uncertainty, and diverse update directions capture information about the model's diversity (Ash et al., 2019). The paper makes no explicit connection to information theory.

Following Proposition 6.1, since BADGE can be seen as using a last-layer approach for the similarity matrix $S[\mathcal{D}^{\mathrm{acq}} \mid \omega^*]$ with hard pseudo-labels, BADGE approximates $\mathrm{I}[\Omega; \{Y_i^{\mathrm{acq}}\} \mid \{x_i^{\mathrm{acq}}\}, \mathcal{D}^{\mathrm{train}}]$ with an uninformative posterior distribution:

> **Proposition 7.2.** *BADGE maximizes an approximation of the EIG with an uninformative prior.*

**Comparison to BatchBALD.** Similarly, BatchBALD (Kirsch et al., 2019) approximates the EIG in the batch acquisition case but by using prediction-space samples. Moreover, BatchBALD uses a greedy approach to select batch candidates instead of sampling via a k-DPP or k-MEANS++.

---

[4]Ash et al. (2021) accidentally writes $\arg\max$ instead of $\arg\min$ in §5.1 in their paper, but c.f. Algorithm 1 with the trace objective. Algorithm 2 & 3 in §B in the appendix use the correct final objective.

As the EIG is submodular, determining the acquisition batch is a submodular optimization problem and, therefore, can be solved by greedy selection with $1 - \frac{1}{e}$ optimality (Nemhauser et al., 1978).

**Research Questions.** Hard pseudo-labels lead to biased estimates. *Would one-sample estimates perform better? And could greedy batch selection work better than sampling via a k-DPP?* This would be closer to the batch acquisition strategy followed by BatchBALD.

### 7.3 SIMILAR (Kothawade et al., 2021) and PRISM (Kothawade et al., 2022)

Based on Iyer et al. (2021), Kothawade et al. (2021) and Kothawade et al. (2022) investigate *submodular active learning* for DNNs: they take an *information function $f$*, which is a non-negative, montone/non-decreasing, submodular function (and then is also subadditive as a consequence):

$$f(A) \geq 0, \qquad \text{(non-negative)}$$
$$f(A) \leq f(B) \text{ for } A \subseteq B, \qquad \text{(monotone)}$$
$$f(A \cup B) \leq f(A) + f(B) - f(A \cap B), \qquad \text{(submodular)}$$
$$f(A \cup B) \leq f(A) + f(B) \qquad \text{(subadditive)}$$

for all $A, B \subseteq \mathcal{D}^{\text{pool}}$ and define a "*submodular conditional gain*" and an "*submodular (conditional) mutual information*" as

$$H_f(A \mid B) := f(A \cup B) - f(B) \qquad (83)$$
$$I_f(A; B) := f(A \cup B) - f(A) - f(B). \qquad (84)$$

For $f(\{x_i\}) = \text{H}[\{Y_i\} \mid \{x_i\}]$, this simply yields the regular information quantities. Hence, Kothawade et al. (2021) and Kothawade et al. (2022) examine other information functions and submodular quantities in the context of active learning: amongst them set covers, graph cuts, facility location, and log determinants (LogDet) of similarity matrices. Like BADGE (Ash et al., 2019), the similarity matrix uses hard pseudo-labels. Like BatchBALD (Kirsch et al., 2019), they use a greedy approach for acquisition (Nemhauser et al., 1978).

Using our results, we immediately see that the LogDet objective, which we can write as $\log \det S[\mathcal{D}^{\text{acq}} \mid \omega^*]$, exactly matches the EIG approximation in §6.2; furthermore, in §D.1, we show that the LogDetMI objective matches an approximation of JEPIG (and similarly, derive the LogDetCMI objective as well):

> **Proposition 7.3.** *The LogDet objective* $\log \det S[\mathcal{D}^{acq} \mid \omega^*]$ *is an approximation of the EIG and the LogDetMI objective*
>
> $$\log \det S[\mathcal{D}^{acq} \mid \omega^*] - \log \det(S[\mathcal{D}^{acq} \mid \omega^*] - S[\mathcal{D}^{acq}; \mathcal{D}^{eval} \mid \omega^*]S[\mathcal{D}^{eval} \mid \omega^*]^{-1} S[\mathcal{D}^{eval}; \mathcal{D}^{acq} \mid \omega^*]) \qquad (85)$$
>
> *is an approximation of a proxy objective for EPIG, where we use $S[\mathcal{D}_1; \mathcal{D}_2 \mid \omega^*]$ to denote the (non-symmetric) similarity matrix between $\mathcal{D}_1$ and $\mathcal{D}_2$.*

Notably, the experimental results for the LogDet-based quantities are reported as among the best in Kothawade et al. (2021) and Kothawade et al. (2022). As such, since the LogDet quantities approximate Shannon's information quantities (which are not explicitly examined in those works), the promising experimental results compared to other submodular information functions support the hypothesis that approximating Shannon's information quantities works well in active learning and active sampling.

**Research Questions.** Similar to BADGE, the scores are biased by using hard pseudo-labels. *Could one-sample estimates perform better?* Furthermore, as LogDetMI is not submodular, *could the approach from BAIT of expanding and shrinking the acquisition batch in a forward and backward pass improve performance here as well?*

### 7.4 Expected Gradient Length

The *Expected Gradient Length (EGL)* (Settles et al., 2007; Settles, 2009) is an acquisition function in active learning and is usually defined for non-Bayesian models. Originally, it was an expectation over the gradient

norm. In more recent literature (Huang et al., 2016), it is introduced using the squared gradient norm:

$$\mathbb{E}_{\mathrm{p}(y^{\mathrm{acq}}|x^{\mathrm{acq}},\omega^*)} \left\| \mathrm{H}'[y^{\mathrm{acq}} \mid x^{\mathrm{acq}}, \omega^*] \right\|^2. \tag{EGL}$$

Using a diagonal approximation of Fisher information, we show in §D.2:

**Proposition 7.4.** *The EIG for a candidate sample $x^{acq}$ approximately lower-bounds the EGL:*

$$2\,\mathrm{I}[\Omega; Y^{acq} \mid x^{acq}] \overset{\approx}{\leq} \mathbb{E}_{\mathrm{p}(y^{acq}|x^{acq},\omega^*)} \left\| \mathrm{H}'[y^{acq} \mid x^{acq}, \omega^*] \right\|^2 + const. \tag{86}$$

### 7.5 Deep Learning on a Data Diet

In active sampling, Paul et al. (2021) use the gradient length of given labeled samples $x, y$ (averaged over multiple training runs) as an acquisition function to select the most informative samples from the training set to speed up training:

$$\mathbb{E}_{\mathrm{q}(\omega)} \left\| \mathrm{H}'[y^{\mathrm{acq}} \mid x^{\mathrm{acq}}, \omega] \right\|^2, \tag{GraNd}$$

which they call the *gradient norm score (GraNd)*. The expectation is taken over the model parameters at initialization or after training for a few epochs—as this is not easily expressed using a posterior distribution, we use $\mathrm{q}(w)$ to denote the distribution.

**Proposition 7.5.** *The IG for a candidate sample $x^{acq}$ approximately lower-bounds the gradient norm score (GraNd) at $\omega^*$ up to a second-order term:*

$$2\,\mathrm{I}[\Omega; y^{acq} \mid x^{acq}] \overset{\approx}{\leq} \mathbb{E}_{\mathrm{q}(\omega)}[\left\| \mathrm{H}'[y^{acq} \mid x^{acq}, \omega] \right\|^2] - \mathbb{E}_{\mathrm{q}(\omega)}[\mathrm{tr}\left( \frac{\nabla^2_\omega \mathrm{p}(y \mid x, \omega)}{\mathrm{p}(y \mid x, \omega)} \right)] + const. \tag{87}$$

The second term might not be negligible. Hence, GraNd (the first term on the left) might deviate from the information gain. *How does the information gain compares to GraNd in practice?*

## 8 Conclusion & Outlook

We have examined Fisher information and Gaussian approximations and have derived weight-space approximations for various information quantities. This has allowed us to connect these information quantities to objectives already used in the literature. Moreover, we can make the following concluding points:

**Last-Layer Approaches.** Methods that only use last-layer Fisher information or similar perform data subset selection on embeddings only, despite feature learning being arguably the most important strength of deep neural networks. However, these approaches can find great use with large pre-trained models, which are only fine-tuned on new data domains (Tran et al., 2022).

**Bias in Hard Pseudo-Labels.** Several methods (BADGE, PRISM, SIMILAR) use hard pseudo-labels for computing the similarity matrices, leading to biased estimates. Can one-sample estimates perform better? Can the equivalent approximations that do not make use of similarity matrices perform better?

**Trace vs** $\log\det$ **Approximations.** We have presented a hierarchy of approximations and bounds. Ablating these approximations along multiple dimensions, including whether to use the trace or $\log\det$ approximation and whether to use a GLM, the GGN approximation or the full Fisher information, could provide interesting insights into what is attainable.

**Batch Acquisition Pathologies.** Approaches that use the matrix trace instead of the log determinant can end up being additive for batch candidates $\{x_i^{\mathrm{acq}}\}$ and, therefore, by definition, cannot take redundancies between batch candidates into account, leading to failures detailed in Kirsch et al. (2019; 2021a). This is an issue for trace approximations of the (E)IG. Does the trace approximation of (JE)PIG handle batch

acquisition pathologies better? Similarly, most information quantities are not submodular, yet we use a greedy algorithm to select the acquisition batches. Is the heuristic proposed by Ash et al. (2021) generally beneficial?

**Weight vs. Prediction Space.** BADGE, BAIT, and the LogDet objectives of PRISM and SIMILAR (Ash et al., 2019; 2021; Kothawade et al., 2021; 2022) approximate information quantities in weight space, while (Batch)BALD, (J)EPIG, and (J)PIG (Houlsby et al., 2011; Kirsch et al., 2019; 2021b; Mindermann et al., 2022) approximate the information quantities in prediction space. Both approaches have their limitations:

Weight-space approaches can suffer from the Gaussian approximation being of low quality: the Laplace approximation only captures the posterior distribution well once it concentrates sufficiently. However, this is unlikely to happen in a low-data regime.

Prediction-space approaches can suffer from a combinatorial explosion as the batch acquisition size increases because prediction configurations have to be enumerated or sampled to approximate the information quantities. In addition, many parameter samples might be needed to obtain low-variance estimates.

Importantly, prediction space approaches also require drawing samples from the posterior distribution but do not estimate the information quantities using the posterior distribution, unlike weight space approaches.

**Informativeness Scores.** Taking a step back, we have seen that a Bayesian perspective using information quantities connects seemingly disparate literature. Although Bayesian methods are often seen as separate from (non-Bayesian) active learning and active sampling, the sometimes fuzzy notion of "informativeness" expressed through various different objectives in non-Bayesian settings collapses to the same couple of information quantities, which were, in principle, already known by Lindley (1956) and MacKay (1992).

## Acknowledgements

The authors would like to thank their anonymous TMLR reviewers for their kind, constructive and helpful feedback during the review process, which has significantly improved this work. We would also like to thank Freddie Bickford Smith, Tom Rainforth, Lisa Schut, and Hugh Goatcher, as well as the members of OATML in general for their feedback at various stages of the project. AK is supported by the UK EPSRC CDT in Autonomous Intelligent Machines and Systems (grant reference EP/L015897/1).

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

## A Fisher Information: Additional Derivations & Proofs

**Proposition 4.2.** *Like observed information, Fisher information is additive:*

$$\mathrm{H}''[\{Y_i\} \mid \{x_i\}, \omega^*] = \sum_i \mathrm{H}''[\{Y_i\} \mid x_i, \omega^*]. \tag{25}$$

*Proof.* This follows immediately from $Y_i \perp\!\!\!\perp Y_j \mid x_i, x_j, \omega^*$ for $i \neq j$ and the additivity of the observed information:

$$\mathrm{H}''[\{Y_i\} \mid \{x_i\}, \omega^*] = \mathbb{E}_{\mathrm{p}(\{y_i\} \mid \{x_i\}, \omega^*)}[\mathrm{H}''[\{y_i\} \mid \{x_i\}, \omega^*]] = \mathbb{E}_{\mathrm{p}(\{y_i\} \mid \{x_i\}, \omega^*)}[\sum_i \mathrm{H}''[y_i \mid x_i, \omega^*]] \tag{88}$$

$$= \sum_i \mathbb{E}_{\mathrm{p}(y_i \mid x_i, \omega^*)}[\mathrm{H}''[y_i \mid x_i, \omega^*]] = \sum_i \mathrm{H}''[y_i \mid x_i, \omega^*]. \tag{89}$$

$\square$

**Proposition 4.3.** *Fisher information is equivalent to:*

$$\mathrm{H}''[Y \mid x, \omega^*] = \mathbb{E}_{\mathrm{p}(y \mid x, \omega^*)}[\mathrm{H}'[y \mid x, \omega^*]^T \; \mathrm{H}'[y \mid x, \omega^*]] = \mathrm{Cov}[\mathrm{H}'[Y \mid x, \omega^*]]. \tag{26}$$

To prove Proposition 4.3, we use the two generally useful lemmas below:

**Lemma A.1.** *For the Jacobian $\mathrm{H}'[y \mid x, \omega^*]$, we have:*

$$\mathrm{H}'[y \mid x, \omega^*] = \nabla_\omega[-\log \mathrm{p}(y \mid x, \omega^*)] = -\frac{\nabla_\omega \mathrm{p}(y \mid x, \omega^*)}{\mathrm{p}(y \mid x, \omega^*)}, \tag{90}$$

*and for the Hessian $\mathrm{H}''[y \mid x, \omega^*]$, we have:*

$$\mathrm{H}''[y \mid x, \omega^*] = \mathrm{H}'[y \mid x, \omega^*]^T \; \mathrm{H}'[y \mid x, \omega^*] - \frac{\nabla_\omega^2 \mathrm{p}(y \mid x, \omega^*)}{\mathrm{p}(y \mid x, \omega^*)}. \tag{91}$$

*Proof.* The result follows immediately from the application of the rules of multivariate calculus. $\square$

**Lemma A.2.** *The following expectations over the model's own predictions vanish:*

$$\mathbb{E}_{\mathrm{p}(y \mid x, \omega^*)}[\mathrm{H}'[y \mid x, \omega^*]] = 0, \tag{92}$$

$$\mathbb{E}_{\mathrm{p}(y \mid x, \omega^*)}\left[\frac{\nabla_\omega^2 \mathrm{p}(y \mid x, \omega^*)}{\mathrm{p}(y \mid x, \omega^*)}\right] = 0. \tag{93}$$

*Proof.* We use the previous equivalences and rewrite the expectations as integral; the results follows:

$$\mathbb{E}_{\mathrm{p}(y \mid x, \omega^*)}[\mathrm{H}'[y \mid x, \omega^*]] = \mathbb{E}_{\mathrm{p}(y \mid x, \omega^*)}[-\nabla_\omega \log \mathrm{p}(y \mid x, \omega^*)] = -\mathbb{E}_{\mathrm{p}(y \mid x, \omega^*)}\left[\frac{\nabla_\omega \mathrm{p}(y \mid x, \omega^*)}{\mathrm{p}(y \mid x, \omega^*)}\right] \tag{94}$$

$$= -\int \nabla_\omega \mathrm{p}(y \mid x, \omega^*)\, dy = -\nabla_\omega \int \mathrm{p}(y \mid x, \omega^*)\, dy = -\nabla_\omega 1 = 0, \tag{95}$$

$$\mathbb{E}_{\mathrm{p}(y \mid x, \omega^*)}\left[\frac{\nabla_\omega^2 \mathrm{p}(y \mid x, \omega^*)}{\mathrm{p}(y \mid x, \omega^*)}\right] = \int \nabla_\omega^2 \mathrm{p}(y \mid x, \omega^*)\, dy = \nabla_\omega^2 \int \mathrm{p}(y \mid x, \omega^*)\, dy = \nabla_\omega^2 1 = 0. \tag{96}$$

$\square$

*Proof of Proposition 4.3.* With the previous lemma, we have the following.

$$\mathrm{Cov}[\mathrm{H}'[Y \mid x, \omega^*]] = \mathbb{E}[\mathrm{H}'[Y \mid x, \omega^*]^T \; \mathrm{H}'[Y \mid x, \omega^*]] - \underbrace{\mathbb{E}[\mathrm{H}'[Y \mid x, \omega^*]^T]}_{=0} \underbrace{\mathbb{E}[\mathrm{H}'[Y \mid x, \omega^*]]}_{=0} \tag{97}$$

$$= \mathbb{E}[\mathrm{H}'[Y \mid x, \omega^*]^T \ \mathrm{H}'[Y \mid x, \omega^*]]. \tag{98}$$

For the expectation over the Hessian, we plug Lemma A.1 into Lemma A.2 and obtain:

$$\mathrm{H}''[Y \mid x, \omega^*] = \mathbb{E}_{\mathrm{p}(y|x,\omega^*)}[\mathrm{H}''[y \mid x, \omega^*]] = \mathbb{E}_{\mathrm{p}(y|x,\omega^*)}\left[\mathrm{H}'[y \mid x, \omega^*]^T \ \mathrm{H}'[y \mid x, \omega^*] - \frac{\nabla^2_\omega \mathrm{p}(y \mid x, \omega^*)}{\mathrm{p}(y \mid x, \omega^*)}\right] \tag{99}$$

$$= \mathbb{E}_{\mathrm{p}(y|x,\omega^*)}[\mathrm{H}'[y \mid x, \omega^*]^T \ \mathrm{H}'[y \mid x, \omega^*]] - 0 = \mathrm{Cov}[\mathrm{H}'[Y \mid x, \omega^*]]. \tag{100}$$

$\square$

## A.1  Special Case: Exponential Family

**Proposition 4.4.** *The Fisher information* $\mathrm{H}''[Y \mid x, \omega^*]$ *for a model* $\mathrm{p}(y \mid \hat{z} = \hat{f}(x; \omega^*))$ *is equivalent to:*

$$\mathrm{H}''[Y \mid x, \omega^*] = \nabla_\omega \hat{f}(x; \omega^*)^T \mathbb{E}_{\mathrm{p}(y|x,\omega^*)}[\nabla^2_{\hat{z}} \mathrm{H}[y \mid \hat{z} = \hat{f}(x; \omega^*)]] \nabla_\omega \hat{f}(x; \omega^*), \tag{27}$$

*where* $\nabla^2_{\hat{z}} \mathrm{H}[y \mid \hat{z} = \hat{f}(x; \omega^*)]$ *is short for* $\nabla^2_{\hat{z}} \mathrm{H}[y \mid \hat{z}]\big|_{\hat{z}=\hat{f}(x;\omega^*)}$.

*Proof.* We apply the second equivalence in Proposition 4.3 twice:

$$\mathrm{H}''[Y \mid x, \omega^*] = \mathrm{Cov}[\mathrm{H}'[Y \mid x, \omega^*]] = \mathrm{Cov}[\nabla_\omega \hat{f}(x; \omega^*)^T \nabla_{\hat{z}} \mathrm{H}[y \mid \hat{z} = \hat{f}(x; \omega^*)] \nabla_\omega \hat{f}(x; \omega^*)] \tag{101}$$

$$= \nabla_\omega \hat{f}(x; \omega^*)^T \mathrm{Cov}[\nabla_{\hat{z}} \mathrm{H}[y \mid \hat{z} = \hat{f}(x; \omega^*)]] \nabla_\omega \hat{f}(x; \omega^*) \tag{102}$$

$$= \nabla_\omega \hat{f}(x; \omega^*)^T \mathbb{E}_{\mathrm{p}(y|x,\omega^*)}[\nabla^2_{\hat{z}} \mathrm{H}[y \mid \hat{z} = \hat{f}(x; \omega^*)]] \nabla_\omega \hat{f}(x; \omega^*) \tag{103}$$

$\square$

## A.2  Special Case: Generalized Linear Models

**Proposition 4.7.** *The observed information* $\mathrm{H}''[y \mid x, \omega^*]$ *of a GLM is independent of* $y$.

$$\mathrm{H}''[y \mid x, \omega^*] = \nabla_\omega \hat{f}(x; \omega^*)^T \nabla^2_{\hat{z}} \mathrm{H}[y \mid \hat{z} = \hat{f}(x; \omega^*)] \nabla_\omega \hat{f}(x; \omega^*) \tag{31}$$

$$= \nabla_\omega \hat{f}(x; \omega^*)^T \nabla^2_{\hat{z}} A(w^T x) \nabla_\omega \hat{f}(x; \omega^*). \tag{32}$$

*Proof.*

$$\mathrm{H}''[y \mid x, \omega^*] \tag{104}$$

$$= \nabla_\omega[\mathrm{H}'[y \mid x, \omega^*]] \tag{105}$$

$$= \nabla_\omega[\nabla_{\hat{z}} \mathrm{H}[y \mid \hat{z} = \hat{f}(x; \omega^*)] \nabla_\omega \hat{f}(x; \omega^*)] \tag{106}$$

$$= \nabla_{\hat{z}} \mathrm{H}[y \mid \hat{z} = \hat{f}(x; \omega^*)] \underbrace{\nabla^2_\omega \hat{f}(x; \omega^*)}_{=\nabla^2_\omega[w^T x]=0} + \nabla_\omega \hat{f}(x; \omega^*)^T \nabla^2_{\hat{z}} \mathrm{H}[y \mid \hat{z} = \hat{f}(x; \omega^*)] \nabla_\omega \hat{f}(x; \omega^*) \tag{107}$$

$$= \nabla_\omega \hat{f}(x; \omega^*)^T \nabla^2_{\hat{z}} A(w^T x) \nabla_\omega \hat{f}(x; \omega^*). \tag{108}$$

$\square$

**Proposition 4.9.** *For a GLM, when* $\hat{f}(x; \omega) : \mathbb{R}^D \to \mathbb{R}^K$, *where* $K$ *is the number of classes (outputs),* $D$ *is the number of input dimensions,* $\omega \in \mathbb{R}^{D \times K}$, *and assuming the parameters are flattened into a single vector for the Jacobian, we have* $\nabla_\omega \hat{f}(x; \omega) = \mathrm{Id}_K \otimes x^T \in \mathbb{R}^{K \times (K \cdot D)}$, *where* $\otimes$ *denotes the Kronecker product, and:*

$$\nabla_\omega \hat{f}(x; \omega^*)^T \nabla^2_{\hat{z}} A(\omega^T x) \nabla_\omega \hat{f}(x; \omega^*) = \nabla^2_{\hat{z}} A(w^T x) \otimes x \, x^T. \tag{35}$$

*Proof.* We begin with a few statements that lead to the conclusion step by step, where $x \in \mathbb{R}^D, A \in \mathbb{R}^{C \times C}, G \in \mathbb{R}^{C \times (C \cdot D)}$:

$$(x\,x^T)_{ij} = x_i\,x_j \tag{109}$$

$$(Id_C \otimes x^T)_{c,d\,D+i} = x_i \cdot \mathbb{1}\{c = d\} \tag{110}$$

$$(G^T\,A\,G)_{ij} = \sum_{k,l} G_{ki}\,A_{kl}\,G_{lj} \tag{111}$$

$$(A \otimes x\,x^T)_{c\,D+i,d\,D+j} = A_{cd}\,x_i\,x_j, \tag{112}$$

$$((Id_C \otimes x^T)^T\,A\,(Id_C \otimes x^T))_{c\,D+i,d\,D+j} = \sum_{k,l}(Id_C \otimes x^T)_{k,c\,D+i}\,A_{kl}\,(Id_C \otimes x^T)_{l,d\,D+j} \tag{113}$$

$$= \sum_{k,l} x_i \cdot \mathbb{1}\{k = c\}\,A_{kl}\,x_j \cdot \mathbb{1}\{l = d\} \tag{114}$$

$$= x_i\,A_{cd}\,x_j \tag{115}$$

$$= (A \otimes x\,x^T)_{c\,D+i,d\,D+j}. \tag{116}$$

$$\implies \nabla_\omega \hat{f}(x;\omega^*)^T\,\nabla_{\hat{z}}^2 A(\omega^T x)\,\nabla_\omega \hat{f}(x;\omega^*) = \nabla_{\hat{z}}^2 A(w^T x) \otimes x\,x^T. \tag{117}$$

$\square$

**Proposition 4.8.** *For a model such that the observed information* $H''[y \mid x, \omega^*]$ *is independent of* $y$, *we have:*

$$H''[Y \mid x, \omega^*] = H''[y^* \mid x, \omega^*] \tag{33}$$

*for any* $y^*$, *and also trivially:*

$$H''[Y \mid x, \omega^*] = \mathbb{E}_{p(y|x)}[H''[y \mid x, \omega^*]]. \tag{34}$$

*Proof.* This follows directly from Proposition 4.5. In particular, we have:

$$H''[Y \mid x, \omega^*] = \mathbb{E}_{p(y|x,\omega^*)}[H''[y \mid x, \omega^*]] = H''[y^* \mid x, \omega^*], \tag{118}$$

where we have fixed $y^*$ to an arbitrary value. $\square$

## B Approximating Information Quantities

### B.1 Approximate Expected Information Gain

**Lemma 5.1.** *For symmetric, positive semidefinite matrices* $A$, *we have (with equality iff* $A = 0$):

$$\log \det(A + Id) \leq \operatorname{tr}(A). \tag{46}$$

*Proof.* When $A$ is positive semidefinite and symmetric, its eigenvalues $(\lambda_i)_i$ are real and nonnegative. Moreover, $A + Id$ has eigenvalues $(\lambda_i + 1)_i$; $\det(A + Id) = \prod_i(\lambda_i + 1)$; and $\operatorname{tr} A = \sum_i \lambda_i$. These properties easily follow from the respective eigenvalue decomposition. Thus, we have:

$$\log \det(A + Id) \leq \log \prod_i(\lambda_i + 1) = \sum_i \log(\lambda_i + 1) \leq \sum_i \lambda_i = \operatorname{tr}(A), \tag{119}$$

where we have used $\log(x + 1) \leq x$ iff equality for $x = 0$. $\square$

**General Case.** In the main text, we only skimmed the general case and mentioned the main assumption. Here, we look at the general case in detail.

For the general case, we need to make strong approximations to be able to pursue a similar derivation. First, we cannot drop the expectation; instead, we note that the log determinant is a concave function on the

positive semidefinite symmetric cone (Cover and Thomas, 1988), and we can use Jensen's inequality on the log determinant term from Equation (41) as follows:

$$\mathbb{E}_{\mathrm{p}(\{y_i^{\mathrm{acq}}\}|\{x_i^{\mathrm{acq}}\})}[\log\det\left(\mathrm{H}''[\{y_i^{\mathrm{acq}}\}\mid\{x_i^{\mathrm{acq}}\},\omega^*]\,\mathrm{H}''[\omega^*]^{-1}+Id)] \tag{120}$$

$$\leq \log\det\left(\mathbb{E}_{\mathrm{p}(\{y_i^{\mathrm{acq}}\}|\{x_i^{\mathrm{acq}}\})}[\mathrm{H}''[\{y_i^{\mathrm{acq}}\}\mid\{x_i^{\mathrm{acq}}\},\omega^*]]\,\mathrm{H}''[\omega^*]^{-1}+Id\right). \tag{121}$$

Second, we need use the following approximation:

$$\mathrm{p}(\{y_i^{\mathrm{acq}}\}\mid\{x_i^{\mathrm{acq}}\})\approx \mathrm{p}(\{y_i^{\mathrm{acq}}\}\mid\{x_i^{\mathrm{acq}}\},\omega^*) \tag{122}$$

to obtain a Fisher information and use its additivity. That is, we obtain:

$$\mathbb{E}_{\mathrm{p}(\{y_i^{\mathrm{acq}}\}|\{x_i^{\mathrm{acq}}\},\omega^*)}[\mathrm{H}''[\{y_i^{\mathrm{acq}}\}\mid\{x_i^{\mathrm{acq}}\},\omega^*]] = \mathrm{H}''[\{Y_i^{\mathrm{acq}}\}\mid\{x_i^{\mathrm{acq}}\},\omega^*] = \sum_i \mathrm{H}''[y^{\mathrm{acq}}{}_i\mid x^{\mathrm{acq}}{}_i,\omega^*]. \tag{123}$$

Plugging all of this together and applying Lemma 5.1, we obtain the same final approximation:

$$\mathrm{I}[\Omega;\{Y_i^{\mathrm{acq}}\}\mid\{x_i^{\mathrm{acq}}\}] = \ldots \approx \tfrac{1}{2}\,\mathbb{E}_{\mathrm{p}(\{y_i^{\mathrm{acq}}\}|\{x_i^{\mathrm{acq}}\})}[\log\det\left(\mathrm{H}''[\{y_i^{\mathrm{acq}}\}\mid\{x_i^{\mathrm{acq}}\},\omega^*]\,\mathrm{H}''[\omega^*]^{-1}+Id)] \tag{124}$$

$$\leq \tfrac{1}{2}\log\det\left(\mathbb{E}_{\mathrm{p}(\{y_i^{\mathrm{acq}}\}|\{x_i^{\mathrm{acq}}\})}[\mathrm{H}''[\{y_i^{\mathrm{acq}}\}\mid\{x_i^{\mathrm{acq}}\},\omega^*]]\,\mathrm{H}''[\omega^*]^{-1}+Id\right) \tag{125}$$

$$\approx \tfrac{1}{2}\log\det\left(\mathbb{E}_{\mathrm{p}(\{y_i^{\mathrm{acq}}\}|\{x_i^{\mathrm{acq}}\},\omega^*)}[\mathrm{H}''[\{y_i^{\mathrm{acq}}\}\mid\{x_i^{\mathrm{acq}}\},\omega^*]]\,\mathrm{H}''[\omega^*]^{-1}+Id\right) \tag{126}$$

$$= \tfrac{1}{2}\log\det\left(\sum_i \mathrm{H}''[Y^{\mathrm{acq}}{}_i\mid x^{\mathrm{acq}}{}_i,\omega^*]\,\mathrm{H}''[\omega^*]^{-1}+Id\right) \tag{127}$$

$$\leq \tfrac{1}{2}\sum_i \mathrm{tr}\left(\mathrm{H}''[Y^{\mathrm{acq}}{}_i\mid x^{\mathrm{acq}}{}_i,\omega^*]\,\mathrm{H}''[\omega^*]^{-1}\right). \tag{128}$$

Unlike in the case of generalized linear models, a stronger assumption was necessary to reach the same result. Alternatively, we could use the GGN approximation, which leads to the same result.

## B.2 Approximate Expected Predicted Information Gain

In the main text, we only briefly referred to not knowing a principled way to arrive at the same result of Proposition 5.4 for the general case. This is because unlike the expected information gain, the Fisher information for an acquisition candidate now lies within an matrix inversion. Even if we used the fact that $\log\det(Id + X\,Y^{-1})$ is concave in $X$ and convex in $Y$, we would end up with:

$$\ldots \tag{129}$$

$$\approx \tfrac{1}{2}\,\mathbb{E}_{\mathrm{p}(y^{\mathrm{eval}},y^{\mathrm{acq}}|x^{\mathrm{eval}},x^{\mathrm{acq}})\,\mathrm{p}(x^{\mathrm{eval}})}[\log\det\left(\mathrm{H}''[y^{\mathrm{eval}}\mid x^{\mathrm{eval}},\omega^*]\,(\mathrm{H}''[y^{\mathrm{acq}}\mid x^{\mathrm{acq}},\omega^*]+\mathrm{H}''[\omega^*])^{-1}+Id)] \tag{130}$$

$$\leq \tfrac{1}{2}\,\mathbb{E}_{\mathrm{p}(y^{\mathrm{acq}}|x^{\mathrm{acq}})}[\log\det\left(\mathbb{E}_{\mathrm{p}(y^{\mathrm{eval}},x^{\mathrm{eval}})}[\mathrm{H}''[y^{\mathrm{eval}}\mid x^{\mathrm{eval}},\omega^*]]\,(\mathrm{H}''[y^{\mathrm{acq}}\mid x^{\mathrm{acq}},\omega^*]+\mathrm{H}''[\omega^*])^{-1}+Id)] \tag{131}$$

$$\geq \tfrac{1}{2}\log\det\left(\mathbb{E}_{\mathrm{p}(y^{\mathrm{eval}},x^{\mathrm{eval}})}[\mathrm{H}''[y^{\mathrm{eval}}\mid x^{\mathrm{eval}},\omega^*]]\,(\mathbb{E}_{\mathrm{p}(y^{\mathrm{acq}}|x^{\mathrm{acq}})}[\mathrm{H}''[y^{\mathrm{acq}}\mid x^{\mathrm{acq}},\omega^*]]+\mathrm{H}''[\omega^*])^{-1}+Id\right) \tag{132}$$

$$= \tfrac{1}{2}\log\det\left(\mathbb{E}_{\mathrm{p}(x^{\mathrm{eval}})}[\mathrm{H}''[Y^{\mathrm{eval}}\mid x^{\mathrm{eval}},\omega^*]]\,(\mathrm{H}''[Y^{\mathrm{acq}}\mid x^{\mathrm{acq}},\omega^*]+\mathrm{H}''[\omega^*])^{-1}+Id\right) \tag{133}$$

$$\leq \tfrac{1}{2}\mathrm{tr}\left(\mathbb{E}_{\mathrm{p}(x^{\mathrm{eval}})}[\mathrm{H}''[Y^{\mathrm{eval}}\mid x^{\mathrm{eval}},\omega^*]]\,(\mathrm{H}''[Y^{\mathrm{acq}}\mid x^{\mathrm{acq}},\omega^*]+\mathrm{H}''[\omega^*])^{-1}\right). \tag{134}$$

Note the $\leq\ldots\geq$, which invalidates the chain. The errors could cancel out, but a principled statement seems hardly possible using this deduction.

## B.3 Approximate Predictive Information Gain

Similarly to Proposition 5.3, we can approximate the predictive information gain. We assume that we have access to an (empirical) distribution $\hat{\mathrm{p}}(x^{\mathrm{eval}},y^{\mathrm{eval}})$:

**Proposition B.1.** *For a generalized linear model (or with the GGN approximation), when we take the expectation over* $\hat{p}(x^{eval}, y^{eval})$*, we have:*

$$\arg\max_{x^{acq}} I[Y^{eval}; y^{acq} \mid X^{eval}, x^{acq}, \mathcal{D}^{train}] = \arg\min_{x^{acq}} I[\Omega; Y^{eval} \mid X^{eval}, y^{acq}, x^{acq}, \mathcal{D}^{train}] \tag{135}$$

*with*

$$I[\Omega; Y^{eval} \mid X^{eval}, y^{acq}, x^{acq}, \mathcal{D}^{train}] \tag{136}$$

$$= \mathbb{E}_{\hat{p}(x^{eval}, y^{eval})} I[\Omega; y^{eval} \mid x^{eval}, y^{acq}, x^{acq}, \mathcal{D}^{train}]$$

$$\approx \mathbb{E}_{\hat{p}(x^{eval}, y^{eval})}\left[\tfrac{1}{2}\log\det\left(H''[y^{eval} \mid x^{eval}, \omega^*]\,(H''[y^{acq} \mid x^{acq}, \omega^*] + H''[\omega^* \mid \mathcal{D}^{train}])^{-1} + Id\right)\right] \tag{137}$$

$$\leq \tfrac{1}{2}\log\det\left(\mathbb{E}_{\hat{p}(x^{eval}, y^{eval})}[H''[y^{eval} \mid x^{eval}, \omega^*]]\,(H''[y^{acq} \mid x^{acq}, \omega^*] + H''[\omega^* \mid \mathcal{D}^{train}])^{-1} + Id\right) \tag{138}$$

$$\leq \tfrac{1}{2}\mathrm{tr}\left(\mathbb{E}_{\hat{p}(x^{eval}, y^{eval})}[H''[y^{eval} \mid x^{eval}, \omega^*]]\,(H''[y^{acq} \mid x^{acq}, \omega^*] + H''[\omega^* \mid \mathcal{D}^{train}])^{-1}\right). \tag{139}$$

All of this follows immediately. Only for the second inequality, we need to use Jensen's inequality and that the log determinant is on the positive semidefinite symmetric cone (Cover and Thomas, 1988). Like for the information gain, there is no difference between having access to labels or not when we have a GLM or use the GGN approximation.

## B.4 Approximate Joint (Expected) Predictive Information Gain

A comparison of EPIG and JEPIG shows that JEPIG does not require an expectation over $\hat{p}(x^{eval})$ but uses a set of *evaluation samples* $\{x_i^{eval}\}$. As such, we can easily adapt Proposition 5.4 to JEPIG and obtain:

**Proposition B.2** (JEPIG)**.** *For a generalized linear model (or with the GGN approximation), we have:*

$$\arg\max_{\{x_i^{acq}\}} I[\{Y_i^{eval}\}; \{Y_i^{acq}\} \mid \{x_i^{eval}\}, \{x_i^{acq}\}, \mathcal{D}^{train}] = \arg\min_{\{x_i^{acq}\}} I[\Omega; \{Y_i^{eval}\} \mid \{x_i^{eval}\}, \{Y_i^{acq}\}, \{x_i^{acq}\}, \mathcal{D}^{train}]$$

$$\tag{140}$$

*with*

$$I[\Omega; \{Y_i^{eval}\} \mid \{x_i^{eval}\}, \{Y_i^{acq}\}, \{x_i^{acq}\}, \mathcal{D}^{train}]$$

$$\approx \tfrac{1}{2}\log\det\left(H''[\{Y_i^{eval}\} \mid \{x_i^{eval}\}, \omega^*]\,(H''[\{Y_i^{acq}\} \mid \{x_i^{acq}\}, \omega^*] + H''[\omega^* \mid \mathcal{D}^{train}])^{-1} + Id\right) \tag{141}$$

$$\leq \tfrac{1}{2}\mathrm{tr}\left(H''[\{Y_i^{eval}\} \mid \{x_i^{eval}\}, \omega^*]\,(H''[\{Y_i^{acq}\} \mid \{x_i^{acq}\}, \omega^*] + H''[\omega^* \mid \mathcal{D}^{train}])^{-1}\right). \tag{142}$$

Similarly, for JPIG, we obtain without relying on the GGN approximation or GLMs:

**Proposition B.3** (JPIG)**.** *We have:*

$$\arg\max_{\{x_i^{acq}\}} I[\{y_i^{eval}\}; \{y_i^{acq}\} \mid \{x_i^{eval}\}, \{x_i^{acq}\}, \mathcal{D}^{train}] = \arg\min_{\{x_i^{acq}\}} I[\Omega; \{y_i^{eval}\} \mid \{x_i^{eval}\}, \{y_i^{acq}\}, \{x_i^{acq}\}, \mathcal{D}^{train}]$$

$$\tag{143}$$

*with*

$$I[\Omega; \{y_i^{eval}\} \mid \{x_i^{eval}\}, \{Y_i^{acq}\}, \{x_i^{acq}\}, \mathcal{D}^{train}]$$

$$\approx \tfrac{1}{2}\log\det\left(H''[\{y_i^{eval}\} \mid \{x_i^{eval}\}, \omega^*]\,(H''[\{y_i^{acq}\} \mid \{x_i^{acq}\}, \omega^*] + H''[\omega^* \mid \mathcal{D}^{train}])^{-1} + Id\right) \tag{144}$$

$$\leq \tfrac{1}{2}\mathrm{tr}\left(H''[\{y_i^{eval}\} \mid \{x_i^{eval}\}, \omega^*]\,(H''[\{y_i^{acq}\} \mid \{x_i^{acq}\}, \omega^*] + H''[\omega^* \mid \mathcal{D}^{train}])^{-1}\right). \tag{145}$$

**Comparison between (E)PIG and J(E)PIG approximations.** As observed information and Fisher information are additive, the difference between the approximations when we have an empirical, that is finite,

evaluation distribution $\hat{p}(x^{\mathrm{eval}})$ with $M$ samples is a factor of $E$ inside the log determinant or trace:

$$\mathbb{E}_{\hat{p}(x^{\mathrm{eval}}, y^{\mathrm{eval}})}[\mathrm{H}''[y^{\mathrm{eval}} \mid x^{\mathrm{eval}}, \omega^*]] = \frac{1}{E} \sum_i \mathrm{H}''[y^{\mathrm{eval}}{}_i \mid x^{\mathrm{eval}}{}_i, \omega^*] = \frac{1}{E} \mathrm{H}''[\{y_i^{\mathrm{eval}}\} \mid \{x_i^{\mathrm{eval}}\}, \omega^*]. \tag{146}$$

For the log determinant, $\frac{1}{E} \log \det(A + Id) \neq \log \det(\frac{1}{E} A + Id)$, but for the trace approximation, we see that both approximations are equal up to a constant factor. For example:

$$\tfrac{1}{2} \mathrm{tr}(\mathbb{E}_{\hat{p}(x^{\mathrm{eval}})}\left[\mathrm{H}''[Y^{\mathrm{eval}} \mid x^{\mathrm{eval}}, \omega^*]\right] (\mathrm{H}''[\{Y_i^{\mathrm{acq}}\} \mid \{x_i^{\mathrm{acq}}\}, \omega^*] + \mathrm{H}''[\omega^* \mid \mathcal{D}^{\mathrm{train}}])^{-1}) \tag{147}$$

$$= \tfrac{1}{2} \mathrm{tr}(\frac{1}{E} \mathrm{H}''[\{Y_i^{\mathrm{eval}}\} \mid \{x_i^{\mathrm{eval}}\}, \omega^*] (\mathrm{H}''[\{Y_i^{\mathrm{acq}}\} \mid \{x_i^{\mathrm{acq}}\}, \omega^*] + \mathrm{H}''[\omega^* \mid \mathcal{D}^{\mathrm{train}}])^{-1}) \tag{148}$$

$$= \tfrac{1}{2E} \mathrm{tr}(\mathrm{H}''[\{Y_i^{\mathrm{eval}}\} \mid \{x_i^{\mathrm{eval}}\}, \omega^*] (\mathrm{H}''[\{Y_i^{\mathrm{acq}}\} \mid \{x_i^{\mathrm{acq}}\}, \omega^*] + \mathrm{H}''[\omega^* \mid \mathcal{D}^{\mathrm{train}}])^{-1}). \tag{149}$$

## C  Similarity Matrices and One-Sample Approximations of the Fisher Information

**Proposition 6.1.** *Given $\mathcal{D}^{train}$, $\{x_i^{acq}\}$ and (sampled) $\{y_i^{acq}\}$, we have for the EIG:*

$$\mathrm{I}[\Omega; \{Y_i^{acq}\} \mid \{x_i^{acq}\}, \mathcal{D}^{train}] \stackrel{\approx}{\leq} \tfrac{1}{2} \log \det \left(S_{\mathrm{H}''[\omega^* \mid \mathcal{D}^{train}]}[\mathcal{D}^{acq} \mid \omega^*] + Id\right) \tag{73}$$

$$\leq \tfrac{1}{2} \mathrm{tr}\, S_{\mathrm{H}''[\omega^* \mid \mathcal{D}^{train}]}[\mathcal{D}^{acq} \mid \omega^*] \tag{74}$$

*Proof.*

$$\mathrm{I}[\Omega; \{Y_i^{\mathrm{acq}}\} \mid \{x_i^{\mathrm{acq}}\}, \mathcal{D}^{\mathrm{train}}] \stackrel{\approx}{\leq} \tfrac{1}{2} \log \det \left(\mathrm{H}''[\{Y_i^{\mathrm{acq}}\} \mid \{x_i^{\mathrm{acq}}\}, \omega^*] \, \mathrm{H}''[\omega^* \mid \mathcal{D}^{\mathrm{train}}]^{-1} + Id\right) \tag{150}$$

$$= \tfrac{1}{2} \log \det \left((\mathrm{H}''[\{Y_i^{\mathrm{acq}}\} \mid \{x_i^{\mathrm{acq}}\}, \omega^*] + \mathrm{H}''[\omega^* \mid \mathcal{D}^{\mathrm{train}}]) \, \mathrm{H}''[\omega^* \mid \mathcal{D}^{\mathrm{train}}]^{-1}\right) \tag{151}$$

$$\approx \tfrac{1}{2} \log \det \left((\hat{\mathrm{H}}'[\mathcal{D}^{\mathrm{acq}} \mid \omega^*]^T \hat{\mathrm{H}}'[\mathcal{D}^{\mathrm{acq}} \mid \omega^*] + \mathrm{H}''[\omega^* \mid \mathcal{D}^{\mathrm{train}}]) \, \mathrm{H}''[\omega^* \mid \mathcal{D}^{\mathrm{train}}]^{-1}\right) \tag{152}$$

$$= \tfrac{1}{2} \log \det \left(\hat{\mathrm{H}}'[\mathcal{D}^{\mathrm{acq}} \mid \omega^*] \, \mathrm{H}''[\omega^* \mid \mathcal{D}^{\mathrm{train}}]^{-1} \hat{\mathrm{H}}'[\mathcal{D}^{\mathrm{acq}} \mid \omega^*]^T + Id\right) \tag{153}$$

$$= \tfrac{1}{2} \log \det \left(S_{\mathrm{H}''[\omega^* \mid \mathcal{D}^{\mathrm{train}}]}[\mathcal{D}^{\mathrm{acq}} \mid \omega^*] + Id\right), \tag{154}$$

where we have used the matrix determinant lemma:

$$\det(AB + M) = \det(BM^{-1}A + Id) \det M. \tag{155}$$

$\square$

**Connection to the Joint (Expected) Predictive Information Gain.** Following eq. (57), JEPIG can be decomposed as the difference between two EIG terms, which we can further divide into three terms that are only conditioned on $\mathcal{D}^{\mathrm{train}}$:

$$\mathrm{I}[\{Y_i^{\mathrm{eval}}\}; Y^{\mathrm{acq}} \mid \{x_i^{\mathrm{eval}}\}, x^{\mathrm{acq}}, \mathcal{D}^{\mathrm{train}}] \tag{156}$$

$$= \mathrm{I}[\Omega; \{Y_i^{\mathrm{eval}}\} \mid \{x_i^{\mathrm{eval}}\}, \mathcal{D}^{\mathrm{train}}] - \mathrm{I}[\Omega; \{Y_i^{\mathrm{eval}}\} \mid \{x_i^{\mathrm{eval}}\}, Y^{\mathrm{acq}}, x^{\mathrm{acq}}, \mathcal{D}^{\mathrm{train}}]$$

$$= \mathrm{I}[\Omega; \{Y_i^{\mathrm{eval}}\} \mid \{x_i^{\mathrm{eval}}\}, \mathcal{D}^{\mathrm{train}}] - \mathrm{I}[\Omega; \{Y_i^{\mathrm{eval}}\}, Y^{\mathrm{acq}} \mid \{x_i^{\mathrm{eval}}\}, x^{\mathrm{acq}}, \mathcal{D}^{\mathrm{train}}] + \mathrm{I}[\Omega; Y^{\mathrm{acq}} \mid x^{\mathrm{acq}}, \mathcal{D}^{\mathrm{train}}] \tag{157}$$

Using Proposition 6.1, we can approximate this as:

$$\mathrm{I}[\{Y_i^{\mathrm{eval}}\}; Y^{\mathrm{acq}} \mid \{x_i^{\mathrm{eval}}\}, x^{\mathrm{acq}}, \mathcal{D}^{\mathrm{train}}] \tag{158}$$

$$\approx \tfrac{1}{2} \log \det \left(S_{\mathrm{H}''[\omega^* \mid \mathcal{D}^{\mathrm{train}}]}[\mathcal{D}^{\mathrm{eval}} \mid \omega^*] + Id\right) - \tfrac{1}{2} \log \det \left(S_{\mathrm{H}''[\omega^* \mid \mathcal{D}^{\mathrm{train}}]}[\mathcal{D}^{\mathrm{acq}}, \mathcal{D}^{\mathrm{eval}} \mid \omega^*] + Id\right) \tag{159}$$

$$+ \tfrac{1}{2} \log \det \left(S_{\mathrm{H}''[\omega^* \mid \mathcal{D}^{\mathrm{train}}]}[\mathcal{D}^{\mathrm{acq}} \mid \omega^*] + Id\right) \tag{160}$$

Furthermore, we can apply the approximation in Proposition 6.2 and find that the $\log \lambda$ terms cancel because $|\mathcal{D}^{\mathrm{acq}}| + |\mathcal{D}^{\mathrm{train}}| = |\mathcal{D}^{\mathrm{acq}} \cup \mathcal{D}^{\mathrm{train}}|$. Taking the limit $\lambda \to 0$, we obtain:

$$\mathrm{I}[\{Y_i^{\mathrm{eval}}\}; Y^{\mathrm{acq}} \mid \{x_i^{\mathrm{eval}}\}, x^{\mathrm{acq}}, \mathcal{D}^{\mathrm{train}}] \tag{161}$$

$$\approx \tfrac{1}{2} \log \det \left( S[\mathcal{D}^{\mathrm{eval}} \mid \omega^*] + \lambda Id \right) - \tfrac{1}{2} \log \det \left( S[\mathcal{D}^{\mathrm{acq}}, \mathcal{D}^{\mathrm{eval}} \mid \omega^*] + \lambda Id \right) + \tfrac{1}{2} \log \det \left( S[\mathcal{D}^{\mathrm{acq}} \mid \omega^*] + \lambda Id \right) \tag{162}$$

$$\to \tfrac{1}{2} \log \det \left( S[\mathcal{D}^{\mathrm{eval}} \mid \omega^*] \right) - \tfrac{1}{2} \log \det \left( S[\mathcal{D}^{\mathrm{acq}}, \mathcal{D}^{\mathrm{eval}} \mid \omega^*] \right) + \tfrac{1}{2} \log \det \left( S[\mathcal{D}^{\mathrm{acq}} \mid \omega^*] \right). \tag{163}$$

Finally, the first term is independent of $\mathcal{D}^{\mathrm{acq}}$, and if we are interested in approximately maximizing JEPIG, we can maximize as proxy objective:

$$\log \det \left( S_{\mathrm{H}''[\omega^* \mid \mathcal{D}^{\mathrm{train}}]}[\mathcal{D}^{\mathrm{acq}} \mid \omega^*] + Id \right) - \log \det \left( S_{\mathrm{H}''[\omega^* \mid \mathcal{D}^{\mathrm{train}}]}[\mathcal{D}^{\mathrm{acq}}, \mathcal{D}^{\mathrm{eval}} \mid \omega^*] + Id \right), \tag{164}$$

or

$$\log \det \left( S[\mathcal{D}^{\mathrm{acq}} \mid \omega^*] \right) - \log \det \left( S[\mathcal{D}^{\mathrm{acq}}, \mathcal{D}^{\mathrm{eval}} \mid \omega^*] \right). \tag{165}$$

## D  Connection to Other Acquisition Functions in the Literature

### D.1  SIMILAR (Kothawade et al., 2021) and PRISM (Kothawade et al., 2022)

**Connection to LogDetMI.** If we apply the Schur decomposition to $\log \det S[\mathcal{D}^{\mathrm{acq}}, \mathcal{D}^{\mathrm{eval}} \mid \omega^*]$ from eq. (165), we obtain the following:

$$\log \det S[\mathcal{D}^{\mathrm{acq}} \mid \omega^*] - \log \det S[\mathcal{D}^{\mathrm{acq}}, \mathcal{D}^{\mathrm{eval}} \mid \omega^*] \tag{166}$$

$$= \log \det S[\mathcal{D}^{\mathrm{acq}} \mid \omega^*] - \log \det S[\mathcal{D}^{\mathrm{eval}} \mid \omega^*] \tag{167}$$

$$- \log \det(S[\mathcal{D}^{\mathrm{acq}} \mid \omega^*] - S[\mathcal{D}^{\mathrm{acq}}; \mathcal{D}^{\mathrm{eval}} \mid \omega^*] S[\mathcal{D}^{\mathrm{eval}} \mid \omega^*]^{-1} S[\mathcal{D}^{\mathrm{eval}}; \mathcal{D}^{\mathrm{acq}} \mid \omega^*]),$$

where $S[\mathcal{D}^{\mathrm{acq}}; \mathcal{D}^{\mathrm{eval}} \mid \omega^*]$ is the nonsymmetric similarity matrix between $\mathcal{D}^{\mathrm{acq}}$ and $\mathcal{D}^{\mathrm{eval}}$ etc.

Dropping $\log \det S[\mathcal{D}^{\mathrm{eval}} \mid \omega^*]$ which is independent of $\mathcal{D}^{\mathrm{acq}}$, we can instead maximize:

$$\log \det S[\mathcal{D}^{\mathrm{acq}} \mid \omega^*] - \log \det(S[\mathcal{D}^{\mathrm{acq}} \mid \omega^*] - S[\mathcal{D}^{\mathrm{acq}}; \mathcal{D}^{\mathrm{eval}} \mid \omega^*] S[\mathcal{D}^{\mathrm{eval}} \mid \omega^*]^{-1} S[\mathcal{D}^{\mathrm{eval}}; \mathcal{D}^{\mathrm{acq}} \mid \omega^*], ) \tag{168}$$

which is exactly the LogDetMI objective of SIMILAR (Kothawade et al., 2021) and PRISM (Kothawade et al., 2022).

We can further rewrite this objective by extracting $S[\mathcal{D}^{\mathrm{acq}} \mid \omega^*]$ from the second term, obtaining:

$$\log \det S[\mathcal{D}^{\mathrm{acq}} \mid \omega^*] - \log \det(S[\mathcal{D}^{\mathrm{acq}} \mid \omega^*] - S[\mathcal{D}^{\mathrm{acq}}; \mathcal{D}^{\mathrm{eval}} \mid \omega^*] S[\mathcal{D}^{\mathrm{eval}} \mid \omega^*]^{-1} S[\mathcal{D}^{\mathrm{eval}}; \mathcal{D}^{\mathrm{acq}} \mid \omega^*]) \tag{169}$$

$$= -\log \det(Id - S[\mathcal{D}^{\mathrm{acq}} \mid \omega^*]^{-1} S[\mathcal{D}^{\mathrm{acq}}; \mathcal{D}^{\mathrm{eval}} \mid \omega^*] S[\mathcal{D}^{\mathrm{eval}} \mid \omega^*]^{-1} S[\mathcal{D}^{\mathrm{eval}}; \mathcal{D}^{\mathrm{acq}} \mid \omega^*]). \tag{170}$$

**Connection to LogDetCMI.** Using information-theoretic decompositions, it is easy to show that:

$$\mathrm{I}[\{Y_i^{\mathrm{eval}}\}; Y^{\mathrm{acq}} \mid \{x_i^{\mathrm{eval}}\}, x^{\mathrm{acq}}, \{Y_i\}, \{x_i\}, \mathcal{D}^{\mathrm{train}}] \tag{171}$$

$$= \mathrm{I}[\{Y_i^{\mathrm{eval}}\}; Y^{\mathrm{acq}}, \{Y_i\} \mid \{x_i^{\mathrm{eval}}\}, x^{\mathrm{acq}}, \{x_i\}, \mathcal{D}^{\mathrm{train}}] - \mathrm{I}[\{Y_i^{\mathrm{eval}}\}; \{Y_i\} \mid \{x_i\}, \mathcal{D}^{\mathrm{train}}]. \tag{172}$$

These are two JEPIG terms, and using above approximations, including (170), leads to the LogDetCMI objective:

$$\log \frac{\det(Id - S[\mathcal{D}^{\mathrm{acq}} \mid \omega^*]^{-1} S[\mathcal{D}^{\mathrm{acq}}; \mathcal{D}^{\mathrm{eval}} \mid \omega^*] S[\mathcal{D}^{\mathrm{eval}} \mid \omega^*]^{-1} S[\mathcal{D}^{\mathrm{eval}}; \mathcal{D}^{\mathrm{acq}} \mid \omega^*])}{\det(Id - S[\mathcal{D}^{\mathrm{acq}}, \mathcal{D} \mid \omega^*]^{-1} S[\mathcal{D}^{\mathrm{acq}}, \mathcal{D}; \mathcal{D}^{\mathrm{eval}} \mid \omega^*] S[\mathcal{D}^{\mathrm{eval}} \mid \omega^*]^{-1} S[\mathcal{D}^{\mathrm{eval}}; \mathcal{D}^{\mathrm{acq}}, \mathcal{D} \mid \omega^*])}. \tag{173}$$

### D.2 Expected Gradient Length

**Proposition 7.4.** *The EIG for a candidate sample $x^{acq}$ approximately lower-bounds the EGL:*

$$2\,\mathrm{I}[\Omega; Y^{acq} \mid x^{acq}] \overset{\approx}{\leq} \mathbb{E}_{\mathrm{p}(y^{acq}\mid x^{acq},\omega^*)} \left\| \mathrm{H}'[y^{acq} \mid x^{acq}, \omega^*] \right\|^2 + const. \tag{86}$$

*Proof.* The EIG is equal to the conditional entropy up to a constant term, via eq. (51) in Proposition 5.2:

$$\mathrm{I}[\Omega; Y^{\mathrm{acq}} \mid x^{\mathrm{acq}}] \overset{\approx}{\leq} \tfrac{1}{2} \log \det \left( \mathrm{H}''[Y^{\mathrm{acq}} \mid x^{\mathrm{acq}}, \omega^*] + \mathrm{H}''[\omega^* \mid \mathcal{D}^{\mathrm{train}}] \right) + \mathrm{const.} \tag{174}$$

We apply a diagonal approximation for the Fisher information and Hessian, noting that the determinant of the diagonal matrix upper-bounds the determinant of the full matrix:

$$\leq \tfrac{1}{2} \log \det \left( \mathrm{H}''_{diag}[Y^{\mathrm{acq}} \mid x^{\mathrm{acq}}, \omega^*] + \mathrm{H}''_{diag}[\omega^* \mid \mathcal{D}^{\mathrm{train}}] \right) + \mathrm{const.} \tag{175}$$

$$= \tfrac{1}{2} \sum_k \log \left( \mathrm{H}''_{diag,kk}[Y^{\mathrm{acq}} \mid x^{\mathrm{acq}}, \omega^*] + \mathrm{H}''_{diag,kk}[\omega^* \mid \mathcal{D}^{\mathrm{train}}] \right) + \mathrm{const.} \tag{176}$$

We use $\log x \leq x - 1$ and that $\mathrm{H}''[\omega^* \mid \mathcal{D}^{\mathrm{train}}]$ is constant:

$$\leq \tfrac{1}{2} \sum_k \left( \mathrm{H}''_{diag,kk}[Y^{\mathrm{acq}} \mid x^{\mathrm{acq}}, \omega^*] + \mathrm{H}''_{diag,kk}[\omega^* \mid \mathcal{D}^{\mathrm{train}}] \right) + \mathrm{const.} \tag{177}$$

$$\leq \tfrac{1}{2} \sum_k \mathrm{H}''_{diag,kk}[Y^{\mathrm{acq}} \mid x^{\mathrm{acq}}, \omega^*] + \mathrm{const.} \tag{178}$$

From Proposition 4.3, we know that the Fisher information is equivalent to the outer product of the Jacobians: $\mathrm{H}''[Y^{\mathrm{acq}} \mid x^{\mathrm{acq}}, \omega^*] = \mathbb{E}_{\mathrm{p}(y^{\mathrm{acq}}\mid x^{\mathrm{acq}},\omega^*)}[\mathrm{H}'[y^{\mathrm{acq}} \mid x^{\mathrm{acq}}, \omega^*] \ \mathrm{H}'[y^{\mathrm{acq}} \mid x^{\mathrm{acq}}, \omega^*]^T]$, and we finally obtain for the diagonal elements:

$$= \tfrac{1}{2} \sum_k \mathbb{E}_{\mathrm{p}(y^{\mathrm{acq}}\mid x^{\mathrm{acq}},\omega^*)} \left[ \mathrm{H}'_k[y^{\mathrm{acq}} \mid x^{\mathrm{acq}}, \omega^*]^2 \right] + \mathrm{const.} \tag{179}$$

$$= \tfrac{1}{2} \mathbb{E}_{\mathrm{p}(y^{\mathrm{acq}}\mid x^{\mathrm{acq}},\omega^*)} \left[ \left\| \mathrm{H}'[y^{\mathrm{acq}} \mid x^{\mathrm{acq}}, \omega^*] \right\|^2 \right] + \mathrm{const.} \tag{180}$$

$$\square$$

### D.3 Deep Learning on a Data Diet

**Proposition 7.5.** *The IG for a candidate sample $x^{acq}$ approximately lower-bounds the gradient norm score (GraNd) at $\omega^*$ up to a second-order term:*

$$2\,\mathrm{I}[\Omega; y^{acq} \mid x^{acq}] \overset{\approx}{\leq} \mathbb{E}_{\mathrm{q}(\omega)}[\left\| \mathrm{H}'[y^{acq} \mid x^{acq}, \omega] \right\|^2] - \mathbb{E}_{\mathrm{q}(\omega)}[\mathrm{tr}\left( \frac{\nabla_\omega^2 \, \mathrm{p}(y \mid x, \omega)}{\mathrm{p}(y \mid x, \omega)} \right)] + const. \tag{87}$$

*Proof.* For any fixed $\omega^*$, the IG is equal to the conditional entropy up to a constant term, via Proposition 5.3:

$$\mathrm{I}[\Omega; Y^{\mathrm{acq}} \mid x^{\mathrm{acq}}] \overset{\approx}{\leq} \tfrac{1}{2} \log \det \left( \mathrm{H}''[y^{\mathrm{acq}} \mid x^{\mathrm{acq}}, \omega^*] + \mathrm{H}''[\omega^* \mid \mathcal{D}^{\mathrm{train}}] \right) + \mathrm{const.} \tag{181}$$

As in the previous proof, we apply a diagonal approximation for the Hessian, noting that the determinant of the diagonal matrix upper-bounds the determinant of the full matrix:

$$\leq \tfrac{1}{2} \log \det \left( \mathrm{H}''_{diag}[y^{\mathrm{acq}} \mid x^{\mathrm{acq}}, \omega^*] + \mathrm{H}''_{diag}[\omega^* \mid \mathcal{D}^{\mathrm{train}}] \right) + \mathrm{const.} \tag{182}$$

$$= \tfrac{1}{2} \sum_k \log \left( \mathrm{H}''_{diag,kk}[y^{\mathrm{acq}} \mid x^{\mathrm{acq}}, \omega^*] + \mathrm{H}''_{diag,kk}[\omega^* \mid \mathcal{D}^{\mathrm{train}}] \right) + \mathrm{const.} \tag{183}$$

Again, we use $\log x \leq x - 1$ and that $\mathrm{H}''[\omega^* \mid \mathcal{D}^{\mathrm{train}}]$ is constant:

$$\leq \tfrac{1}{2} \sum_k \left( \mathrm{H}''_{diag,kk}[y^{\mathrm{acq}} \mid x^{\mathrm{acq}}, \omega^*] + \mathrm{H}''_{diag,kk}[\omega^* \mid \mathcal{D}^{\mathrm{train}}] \right) + \mathrm{const.} \tag{184}$$

Table 2: *Spearman Rank Correlation of Prediction-Space and Weight-Space Estimates.* BALD and EPIG are both strongly positively rank-correlated with each other. The weight-space approximations are strongly rank-correlated with the prediction-space approximations, but the weight-space approximations are less accurate than the prediction-space approximations. Note that we have reversed the ordering for the proxy objectives for JEPIG and EPIG as they are minimized while EPIG is maximized.

|  | BALD (Prediction) | EIG (LogDet) | EIG (Trace) | EPIG (Prediction) | EPIG (LogDet) | JEPIG (LogDet) | (J)EPIG (Trace) |
|---|---|---|---|---|---|---|---|
| BALD (Prediction) | 1.000 | 0.955 | 0.940 | 0.984 | 0.948 | 0.955 | 0.927 |
| EPIG (Prediction) | 0.984 | 0.918 | 0.897 | 1.000 | 0.918 | 0.918 | 0.903 |

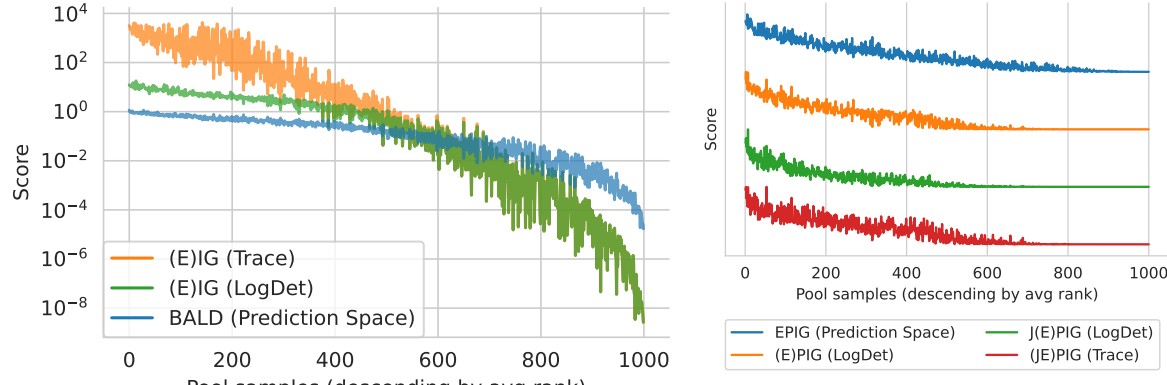

Figure 2: *EIG Approximations.* Trace and log det approximations match for small scores (because of Lemma 5.1). They diverge for large scores. Qualitatively, the order matches the prediction-space approximation using BALD with MC dropout.

Figure 3: *(J)EPIG Approximations (Normalized).* The scores match qualitatively. Note we have reversed the ordering for the proxy objectives for JEPIG and EPIG as they are minimized while EPIG is maximized.

$$\leq \tfrac{1}{2} \sum_k \mathrm{H}''_{diag,kk}[y^{\mathrm{acq}} \mid x^{\mathrm{acq}}, \omega^*] + \mathrm{const.} \tag{185}$$

From Lemma A.1, we know that the Hessian is equivalent to the outer product of the Jacobians plus a second-order term: $\mathrm{H}''[y^{\mathrm{acq}} \mid x^{\mathrm{acq}}, \omega^*] = \mathrm{H}'[y^{\mathrm{acq}} \mid x^{\mathrm{acq}}, \omega^*]\,\mathrm{H}'[y^{\mathrm{acq}} \mid x^{\mathrm{acq}}, \omega^*]^T - \frac{\nabla_\omega^2\, \mathrm{p}(y^{\mathrm{acq}}|x^{\mathrm{acq}},\omega^*)}{\mathrm{p}(y^{\mathrm{acq}}|x^{\mathrm{acq}},\omega^*)}$, and we finally obtain for the diagonal elements:

$$= \tfrac{1}{2} \sum_k \mathrm{H}'_k[y^{\mathrm{acq}} \mid x^{\mathrm{acq}}, \omega^*]^2 - \tfrac{1}{2} \mathrm{tr}\left( \frac{\nabla_\omega^2\, \mathrm{p}(y^{\mathrm{acq}} \mid x^{\mathrm{acq}}, \omega^*)}{\mathrm{p}(y^{\mathrm{acq}} \mid x^{\mathrm{acq}}, \omega^*)} \right) + \mathrm{const.} \tag{186}$$

$$= \tfrac{1}{2} \left\| \mathrm{H}'[y^{\mathrm{acq}} \mid x^{\mathrm{acq}}, \omega^*] \right\|^2 - \tfrac{1}{2} \mathrm{tr}\left( \frac{\nabla_\omega^2\, \mathrm{p}(y^{\mathrm{acq}} \mid x^{\mathrm{acq}}, \omega^*)}{\mathrm{p}(y^{\mathrm{acq}} \mid x^{\mathrm{acq}}, \omega^*)} \right) + \mathrm{const.} \tag{187}$$

Taking an expectation over $\omega^* \sim \mathrm{q}(\omega)$ yields the statement. $\qquad\square$

# E   Preliminary Empirical Comparison of Information Quantity Approximations

The following section describes an initial empirical evaluation[5] of the bounds from §5 on MNIST. We train a model on a subset of MNIST and compare the approximations of EIG and EPIG in weight space to BALD and EPIG computed in prediction space. We use a last-layer approach (GLM) which means that active sampling and active learning approximations are equivalent. We do not attempt to estimate JEPIG in prediction space.

---

[5]Code at: https://github.com/BlackHC/2208.00549

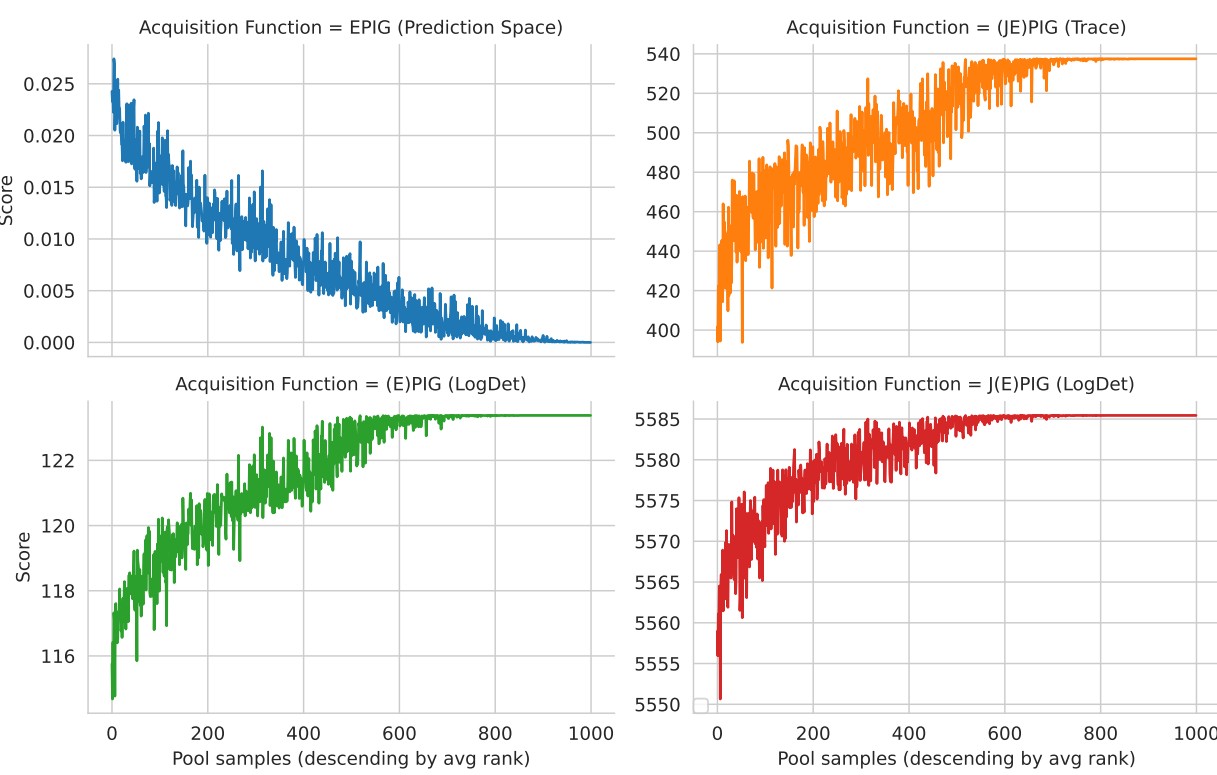

Figure 4: *(J)EPIG Approximations.* The scores match quantitatively. Note the proxy objectives for JEPIG and EPIG are minimized while EPIG is maximized. The value ranges are off by a lot: the true EPIG score is upper bounded by $\log K \approx 2.3\, nats$.

**Setup.** We train a BNN using MC dropout on 80 randomly selected training samples from MNIST, achieving 83% accuracy. The model architecture follows the one described in Kirsch et al. (2019). We use 100 Monte-Carlo dropout samples (Gal and Ghahramani, 2016) to compute the prediction-space estimates. For EPIG, we sample the evaluation set from the remaining training set (20000 samples). We randomly select 1000 samples from the training set as pool set. We compute BALD and EPIG in prediction space as described in Gal et al. (2017) and Kirsch et al. (2021b). For the weight-space approximations, we use a last-layer approximation—we thus have a GLM. For the implementation, we use PyTorch (Paszke et al., 2019) and the laplace-torch library (Daxberger et al., 2021).

We chose 80 samples and 83% accuracy as the accuracy trajectory of BALD and EPIG is steep at this point, see e.g. Kirsch et al. (2019), and thus we expect a wider range of scores.

**Results.** In Figure 2, we see a comparison of BALD with the approximations in eq. (48) and eq. (49). Not shown is eq. (51), which performs like eq. (48) (up to a constant). In Figure 3 and Figure 4, we show a comparison of EPIG with the approximations in Equation (65), Equation (66), and Equation (141). Figure 3 shows normalized scores individually as the score ranges are very different.

Importantly, while the prediction-space scores (BALD and EPIG) have valid scores as the EIG/BALD and EPIG scores of a sample are bounded by the $\log K$, the weight-space scores are not valid. As such, they only provide rough estimates of the information quantities.

However, as we see in Table 2, the Spearman rank correlation coefficients between the weight-space and prediction-space scores are very high. Thus, while the scores themselves are not good estimates, their order seems informative, and this is what matters for selecting acquistion samples in data subset selection.

**Future Work.** The quality of these approximations needs further verification using more complex models and datasets. Comparisons in active learning and active sampling experiments are also necessary to validate the usefulness of these approximations. However, given the connections shown in §7, we expect that the approximations will be useful in these settings as well.

