# OpenReview forum: "Unifying Approaches in Active Learning and Active Sampling via Fisher Information and Information-Theoretic Quantities"
_TMLR — Accepted by TMLR_

### Review · Reviewer_xAk1 · 2022-08-25

**Summary Of Contributions:**

This paper connects a variety of existing methods used in active learning for sample selection and provides detailed background information on existing techniques in the field. It focuses on using methods of approximating the Fisher information and Hessian to come up with new upper bounds for common sampling criteria (EIG, IG, EPIG, JEPIG, PIG, JPIG) and connects previously introduced information quantities to each other.

**Broader Impact Concerns:**

No broader impact concerns.

**Requested Changes:**

Please address the concerns and questions in the weaknesses section.

**Strengths And Weaknesses:**

Strengths:
- Establishes upper bounds on common information theoretic quantities of interest
- Provides mathematical examples for special cases (GLMs, exponential families)
- Thorough background section and relates previous works to one another

Weaknesses/Questions:
- Presentation of the background was not easy for me to read. At times I was unsure of what was new work vs existing work. More room for motivation, comparison of effectiveness of these different methods.
- No mention of implementation details or whether relevant software packages already exist.
- Limited analysis of the quality of the bounds. How “good” are these approximations in practice? Further, how good is the bound?
- The empirical Fisher information matrix has been found to not always give good second-order information (in the Kunster 2019 paper that is referenced). How does this affect the upper bounds?
- Just above section 5.2:  “The log determinant of the Fisher information version might well capture these dependencies.”
Is there more to support this conjecture (intuition, theory)? How could we check this? Does this necessarily mean a tighter bound?

---

> ### Author Response · Authors · 2022-09-17
> **Response to Reviewer xAk1**
>
> Dear Reviewer xAk1,
>
> Thank you so much for your review!
>
> The contribution of our paper is to comprehensively examine approximations of information quantities using Fisher information and show their connections to existing literature. We explicitly limit the significance and impact of our work by not including empirical evaluations of the bounds and leaving those to future work.
>
> Is this limitation of the paper’s significance and impact an obstacle to your acceptance?
>
> We have expanded the paper significantly based on your feedback and have uploaded a major revision:
>
> > Presentation of the background was not easy for me to read. At times I was unsure of what was new work vs existing work. More room for motivation, comparison of effectiveness of these different methods.
>
> We have expanded the introduction to make the contributions clearer. In particular, we clarify that in §3 and §4, our (small) contribution is summarizing results and introducing a unified notation. §5 provides a comprehensive overview that also contrasts the approximations in a newly added Figure 1 now. §6 is novel in its connection between Fisher/observed information and similarity matrices.
>
> > Limited analysis of the quality of the bounds. How “good” are these approximations in practice? Further, how good is the bound?
>
> Generally, this paper is not meant to validate the bounds empirically but only relate them to each other. The approximations seem to perform well, however. BADGE and BAIT both compute such approximations and are SOTA in AL. We intend to run ablations for the hierarchy of approximations, together with reproductions of the other literature, in an empirical follow-up work of similar scope.
>
> > No mention of implementation details or whether relevant software packages already exist.
>
> This paper does not have experiments and implementations, and thus there are no implementation details.
>
> Generally, for Laplace approximations: https://aleximmer.github.io/Laplace/ and https://github.com/kazukiosawa/asdfghjkl
>
> Specific (re)implementations of the referenced prior work also exist for [(Batch)BALD](https://github.com/BlackHC/batchbald_redux), [BADGE, BAIT](https://github.com/JordanAsh/badge), [PRISM & SIMILAR](https://github.com/decile-team/distil/).
>
> Given that we do not include experiments and the software ecosystem changes rapidly, we have not included references to these packages in this revision.
>
> > The empirical Fisher information matrix has been found not always to give good second-order information (in the Kunster 2019 paper that is referenced). How does this affect the upper bounds?
>
> The empirical Fisher information in Kunstner is just the observed information (as defined in this paper) and might not be positive semidefinite (unlike the Fisher information). This leads to issues when optimizing the model using natural gradient methods. This could also lead to issues with log determinant approximations. Without running extensive experiments, we do not want to hypothesize much.
>
> We want to examine the empirical quality of these approximations and such issues in follow-up work as we think proper treatment will require space and time.
>
> > Just above section 5.2: “The log determinant of the Fisher information version might well capture these dependencies.” Is there more to support this conjecture (intuition, theory)? How could we check this? Does this necessarily mean a tighter bound?
>
> A simple experimental setting would be to compare active learning performance on datasets with duplicated data (e.g. RepeatedMNIST from Kirsch et al, 2019). The trace objectives for the EIG will underperform because top-k acquisition underperforms.
>
> Our hypothesis would be that the log determinant version captures redundancies, so the log determinant approximation of the EIG ought not to underperform when using a greedy algorithm and perform similarly to BatchBALD as both approximate the EIG.
>
> We apologize for the delay in responding: updating the paper took longer than expected. We hope the update is to your liking, and we hope that the fact that our paper limits its significance and impact by not including empirical evaluations at this point is acceptable to you.
>
> Thank you so much for your review, and please let us know what you think,\
>  the Authors

---

> ### Author Response · Authors · 2022-09-20
> **Appendix E: Preliminary Empirical Comparison of Information Quantity Approximations**
>
> Dear Reviewer xAk1,
>
> we have added an additional appendix E that compares the weight-space approximations from this paper to prediction-space approximations (computed using MC dropout, following Gal et al, 2017, etc) for a fixed model on MNIST trained to 83% accuracy with 80 training samples.
>
> While the approximations are quantitatively different, qualitatively they perform very similarly to the prediction space ones (which are good estimators for individual acquisition but don't scale to larger acquisition sizes). In the appendix, we also compute the Spearman rank correlation coefficients and find them to be very high (>0.90). The order of the scores is more important for acquisitions in active learning and active sampling.
>
> We leave full active learning and active sampling experiments as future work because that would expand the scope a lot.
>
> Please let us know what you think.
>
> Thanks so much and best wishes,\
>  the Authors
>
> ---
> ![EIG Comparison](https://i.postimg.cc/RFGPpFYJ/bald-eig-log-det-trace-comparison.png)
> **Figure 2: EIG Approximations. Trace and log det approximations match for small scores (because of Lemma 5.1). They diverge for large scores. Qualitatively, the order matches the prediction-space approximation using BALD with MC dropout.**
>
> ![EPIG Comparison](https://i.postimg.cc/RVt7NhT1/epig-log-det-trace-jepig-qualitative-comparison.png)]
> **Figure 3: (J)EPIG Approximations (Normalized). The scores match qualitatively. Note we have reversed the ordering for the proxy objectives for JEPIG and EPIG as they are minimized while EPIG is maximized.**

---

### Review · Reviewer_7mfr · 2022-08-25

**Summary Of Contributions:**

Overall, the primary goal of the work appears to be to argue that a wide array of existing acquisition functions used in active learning and active sampling (BAIT, ActiveSetSelection, BADGE, SIMILAR, PRISM, EGL, and GraND) can be understood as various approximations to the more traditional information theoretical acquisition function of information gain (IG, along with its variants EIG, PIG, EPIG, JPIG, JEPIG).  In particular, these existing acquisition functions are motivated as approximations of information gain powered by a second order, laplace-like, approximation of the posterior distribution, which motivates approximating information gain by means of the properties of the Fisher information matrix.

As a secondary contribution, the paper includes a wide array of useful mathematical summaries and results, including the same fisher powered approximations of classic acquisition functions.

**Broader Impact Concerns:**

I don't have any broader impact concerns about the paper, it is very abstract and theoretical in nature.

**Requested Changes:**

Overall, I hope its clear that my biggest request would be a fairly substantial edit to the prose of the paper.  A lot more signposting, reflection, repetition and handholding would help the reader a lot.  I think it would substantially strengthen the impact of the paper and the size of its audience.  I'm not entirely sure if this is or should be a requirement for my acceptance, but obviously, if taken seriously I would have no hesitation recommending the paper for acceptance.

Some other lesser suggestions:

In Equation 8, some brackets would help disambiguate the action of the operators.
$$ \nabla_\omega \log p(\omega^*) (\omega - \omega^*) = \left( \nabla_\omega \log p(\omega) \right) (\omega - \omega^*) \neq \nabla_\omega[ \log p(\omega^*) (\omega - \omega^*)] $$
or if we want to avoid additional brackets just swap the order of the terms
$$ (\omega - \omega^*) \nabla_\omega \log p(\omega^*) $$
or minimally a footnote about your binding convention for $\nabla_\omega$.  Its a minor problem and the reader can figure it out, but overall one of the strengths of the paper is its careful and concise notation so the ambiguity here is a bit jarring.

I think there is a spurious extra comma in equation 17 $H(\mathcal D | , \omega^*) \to H(\mathcal D | \omega^*)$

The "empirical" fisher note, as worded I think does more harm than good.  The notation is very suggestive of the other "empirical" fisher as there isn't any distinction between the model sampled labels ${y_i}$ and observed empirical data points $\{x_i\}$ except in the functional form on the far left.  This might be a good time to use hats or something for the xs?  I'm not really sure, but I find the notation in 22 very confusing and it tripped me up on first read.  Perhaps start with definition of the per instance Fisher in 24 and then define the Fisher in 4.1 with the sum in 23?

I think 25 would be aided with the intermediate equality $-\nabla_\omega \log p(y|x,\omega^*)$.

But as a broader question why are Lemma 4.4 and 4.5 included in the main body?  Are they useful outside of the proof in appendix A?

Proposition 4.7, typo, I think you want to say exponential family distribution not exponential distribution.
and again on the line below eqn 30.

In the yellow box in the conclusion, please add the acronyms for the acquisition functions along with the references.

minor nits, where they are defined, the (joint) expected predictive information gain (EPIG and JEPIG respectively), you should change the order in which they appear in the parentheses, no?


**Strengths And Weaknesses:**

Overall, I think the paper delivers on its aims.  It does show us how a wide array of existing acquisition functions can be understood to be approximations to information gain.  The paper also contains a lot of useful mathematical manipulations and lemmas and could serve as a  useful reference for people interested in these sorts of things.

However, I feel as though the paper could be greatly improved with some editing.  As it is written, I found the paper very difficult to read.  In general, the paper lacks signposting and hand holding.  It isn't until page 11 that we really get some payoff from what forms the bulk of the paper, which is some rather detailed mathematical manipulations.  And even then, what feels like the primary selling point of the paper, actually demonstrating how a wide class of acquisition functions can be understood through the same lens, feels quite rushed.

The math is good and results can be interesting, but at present the paper really stretches the reader's good will.  As a reviewer I had a very strong incentive to read the paper carefully and many times.  Perhaps people who work very closely with acquisition functions, active learning and active sampling would be similarly motivated to work through it, but I can't help but feel as though the paper could attract a much wider audience with some edits to the prose focused on saying more of the quiet parts out loud.

As a reader, I would appreciate a lot more hand-holding, signposting and direction. Through much of the paper I wasn't quite sure what we were doing, why we were doing it or why I should care.  Then once the result was generated, I felt abandoned when the paper moved on to the next section without helping me to appreciate what we had just accomplished.  Overall it really felt like a lot was left unsaid.

As a small microcosm of the sort of thing I'm trying to get across, at the bottom of page 2 we have a bolded section in the Background and Setting section that tells us that Transductive Objectives are when an objective uses additional data to guide acquisitions.  Being right at the beginning of the paper, and having a dedicated bolded section, as a reader, I was expecting transductive objectives to play a dominant and immediate role.  However, it isn't until page 9, 7 pages later that we'll see the word transductive appear again, where it is essentially redefined for the reader.  In total it only appears 4 additional times throughout the whole paper.  This strikes me as a mismanagement of the readers attention and valuable real estate on the second page of the paper.

Another example of the same sort: the active learning and active sampling sections immediately afterwards.  Here, the goal is to define IG, EIG, PIG, EPIG, JPIG, and JEPIG.  After my first reading of the section I was feeling very overwhelmed, as despite all 6 of these quantities being very closely related, my initial read left me only knowing that 6 different things have been defined and I was worried as a reader I would be expected to now be able to maintain the distinctions betwixt them. They seem to, rather simply, span the set of three independent choices: a difference in whether the acquisition labels are observed or inferred ("expected"), the difference between being information defined in weight or prediction space ("predictive") and the difference between being defined per example or jointly for a batch ("joint"). Not being familiar with these different acquisition functions before reading the paper, this was the sort of work I had to do as a reader that I would have instead appreciated the paper did for me.  Section 2 *defines* what these different acquisition functions are, but doesn't help me *understand* what they are and how they relate.

I understand that styles differ but right now, the paper puts all of the cognitive load out front, Section 2 defines many terms before we know why we need them. Section 3 then takes us through the second order approximation of the posterior and associated entropy but prior to any motivation for why we might be interested in such a thing.  It wasn't until my second read through the paper that I appreciated that the second order approximation was crucial for establishing the links to the other acquisition functions.  Then at the same time, after working through Section 3 for the first time, we end with Preposition 3.5 and then we immediately move on to Section 4.  I feel as though with just a few sentences at the beginning of Section 3 telling the reader what we are about to do and why, and then a couple more sentences at the end of Section 3 spelling out for the reader what was just accomplished and why it is interesting, readers on their first passes through the paper, will have a much improved experience.

In the same sort of spirit, overall, I feel a weakness of the paper is a failure to elucidate its own results.  For example, in Section 5 we work out formulas for EIG, IG, EPIG and in the approximate second order and exponential family approximations the paper explores.  However, I'm not sure what the payoff is.  For example we generate equation 48, but as a reader, I'm not sure why we care.  Is this formula easier to evaluate than other approximations of EIG?  Does this formula make it easier to understand some properties of EIG or reason about the accuracy of the approximation?  Is this something that will be useful later in the paper, or useful later in my life?  Really, I was hoping for a lot more sections like the, well done, "Batch Acquisition Pathologies" or the "Comparison to EIG" sections.  Here the paper takes a moment to reflect on what has been done and help make connections in the reader's mind.  Please add more sections like those throughout.  Please let the reader stand on your shoulders more.

Reaching the end of section 5, we are told that the same kinds of approximations for JEPIG, PIG and JPIG are in the appendices.  This again feels like a bit of a rug pull.  As a reader I'm left wondering whether, either, the JEPIG, PIG and JPIG approximations are important and I've been robbed of them in the main text due to space, or more pessimistically, left to wonder if perhaps the approximations for EIG, IG, and EPIG are also perhaps not so important if their brethren could so easily be disposed of.  Why should we care about these approximations, what do they teach us?

Having laid all of the groundwork, in Section 7 the paper is able to very quickly establish links between many existing acquisition functions BAIT, ActiveSetSelection, BADGE, SIMILAR, PRISM, EGL, and GraND. We are now capable of seeing how they arise from various second order approximations to various forms of the information gain.  The paper does accomplish its goal of establishing these links, to its credit, and given the extensive space dedicated to the mathematical setup, the associations can be made rather quickly: to its strength.  However, again, I have to admit, as a reader, after my first few reads of the paper, I felt robbed.

Section 7 goes by very quickly, and being terse it assumes a high level of familiarity with the existing literature, as well as a great deal of familiarity with the previous sections of the paper.  I understand the triumph that comes from finding a unified perspective or unique lens by which you can suddenly make sense of whole swath of existing work and I believe such perspectives should be shared and cherished.  At the same time, as authors you need to realize that this realization likely took a good amount of time and effort on your part to build up familiarity with the new framework before they clicked as being so obvious.  For readers of this paper, the paper itself is the only introduction they will have had to this new framework, and so I think you owe the readers a great deal more explanation, a more detailed exposition on the relationships and power the new framework provides.

I also would very much appreciate a more thorough overview of what BAIT, ActiveSetSelection, BADGE, SIMILAR, PRISM, EGL, and GraND themselves are or do.  I realize space is always tight, however if the one of the primary goals of this paper is to leave readers understanding how all these various acquisition functions relate, I feel more time spent explaining what these various acquisition functions *are* even if traded for more space for the mathematical details (which can always be relegated to an appendix), the better as far as adding rhetorical punch goes.  Honestly, I'd suggest considering replacing the current background section 2 with a background section that gave an overview of these acquisition functions.  I feel that would do a better job to motivate our task of unifying them.

The Conclusion, in Section 8 does a good job of letting the readers stand on the authors shoulders.  It helps us understand from a high level the implications of what we have just worked through.  I wish only there was more of it, that these sorts of high level reflections on the work being done in the paper would appear repeatedly and throughout the paper itself.  I'm left with many questions.  Some of these methods approximate information quantities in weight space while others approximate in prediction space, so what does that tell us?  Does that effect their accuracy? Their utility? Their deficiencies? Should we prefer one over the other?  I now understand that these things differ, but again, I'm not sure why I should care, I'm not sure how knowing this helps me.

Overall, I believe the paper does what it purports to do, and that there is an audience for its findings.  At the same time I believe it has a lot of potential to reach a much larger audience and provide a much greater benefit to its readers which some edits to the prose.

---

> ### Author Response · Authors · 2022-09-17
> **Response to Reviewer 7mfr**
>
> Dear Reviewer 7mfr,
>
> Thank you very much for the kind and extensive review with many great suggestions. We are very happy that you think that the paper delivers on its aims and that it could serve as a useful reference. We are very grateful for the thorough review that shows that you have gone through the paper in great detail.
>
> Based on your input, we have made substantiative changes and expanded the paper:
>
> ### A lot more hand-holding, signposting, and direction
>
> We have expanded the introduction and all sections to contain more explanations about how the different sections and results fit into the bigger picture.
>
> ### Understanding the different acquisition functions
>
> We have added Table 1 which puts the different acquisition functions into context along the dimensions you have also recognized. We hope that this helps clarify the context of the acquisition functions.
>
> ### Purpose of the approximations
>
> To be of more use as a reference, we have added a diagram (Figure 1) that comprehensively shows the connection of the main approximations that use the log determinant and matrix trace.
> We have changed the notation of the Fisher information to $H''$ to be in line with the observed information and the notation for information quantities: $$H’’[Y \mid x,\omega^*]=\mathbb{E}_{p(y \mid x, \omega^*)} H’’[y\mid x, \omega^*].$$
>
> This change makes many statements more intuitive and makes it easier to see the similarity between active learning and active sampling approaches.
>
> Enumerating different ways of approximating information quantities makes it easier to spot them “in the wild.” For example, we have added a paragraph to BAIT in §7 that shows that one of the ablations from the original paper actually maximizes the negative EIG (ie minimizes the EIG) and is thus trivially different from BAIT. The ablation is meant to compare the trace approximation with the log determinant approximation, yet the authors forget “+ Id” in the log determinant term.
>
> By providing a more comprehensive overview, we hope that similar connections in other literature will become more easily visible to the readers.
>
> ### §7 is terse
>
> We have expanded §7 to give more details about the approaches in the existing literature and include possible research questions.
>
> ### Expand the conclusion
>
> We have expanded the conclusion and trade-offs between weight- and prediction-space methods. We also enumerate possible future research questions.
>
> ###  Lesser suggestions
>
> We have updated the paper according to your lesser suggestions.
>
> We have added about 5.5 pages in total. We apologize for the delay in responding: updating the paper took longer than expected. We hope the update is to your liking.
>
> Thank you so much for your review, and please let us know any other improvements we could make,\
> the Authors

---

### Review · Reviewer_do8M · 2022-09-04

**Summary Of Contributions:**

Contribution-wise, the authors' main claim is the derivation of a taxonomy of DSS approaches based on information theoretic quantities.

More in details, the authors begin by introducing two fundamental quantities related to the entropy of a posterior: its hessian (for taylor expansion) and the fisher information (sec 2,3,4). Then (sec 5,6) the authors show how some of the most popular and generic active learning acquisition measures (EIG and [J]PEIG) can be approximated using these quantities, and provide some insight on the behavior of these approximation under different assumptions. Finally (sec 7) the authors try to interpret several recent contributions in the active selection literature (BAIT, BADGE, EGL, ...) as different approximation approaches to these information measures, in an effort to unify their presentation and provide a general understanding the choices of each different approach when constructing the approximation.



**Broader Impact Concerns:**

No broader impact needed

**Requested Changes:**

W1) This is more on the impact side and future work, i.e. having a better/stronger framework

W2) Again on the impact side, but more immediate. It would be good to consider at least one new intermediate approach in detail, either to show that it might have new strengths, or to motivate a negative result of why it would fail and avoid other people from wasting time on it in the future.

W3) Introduce references to sub-modularity

W4) Easier said than done before trying to write the paper myself. Maybe spread section 7 across Sec. 5/6 to be closer to the logical path, or include a proof graph as a visual aid to navigate the derivation.

**Strengths And Weaknesses:**

Overall the paper has two main strength points:
S1) A good single exposition of many steps that are usually "left to the reader" in the original papers, in the form of a comprehensive chain of prepositions
S2) Insights in the "true" motivation behind model choice in recent literature. As an example they show why GLM and exponential models, despite being restrictive, result in much simpler information measures and easier approximations. This might be well known to experts in the field but sheds some light for someone that just approached it.

The main weaknesses are:
W1) Despite the attempt to unify all the approaches, many of these are only a loose fit in the framework and the authors have to fall back to more vacuous statements such as "approximately lower bounds". Similarly, many of the original methods were already originally derived starting from an information measure perspective but had to cut corners and implement heuristics to become practically efficient presented   (e.g. BADGE replacing k-dpps with k-means++). This makes fitting them in a clean framework (including the one presented by the authors) more difficult.

W2) Creating a new unified framework can have various benefits. One is simplifying exposition and understanding, which the authors achieve. Another is using the more general formulation to obtain novel insights, which the authors only partially achieve (e.g. last layer and batching pathologies were known, but the authors give a stronger and cleaner understanding). Finally, using the more general framework to explore new algorithmic design space that interpolates between existing special cases, which the authors do not attempt.

W3) The work is narrowly tailored in an information theoretic/bayesian perspective, but many of the comments regarding entropy and information gain approximation can also be tackled from a sub-modular function approximation perspective. It would be important to at least introduce this to reader as a potential separate path when looking for unifying frameworks.

W4) A minor comment would be that on a 12 page paper, the "meat" of it (i.e. tying the framework to the actual algorithms) takes place in only the last 2 pages (section 7). This leaves the reader a bit confused trying to backtrack the logic flow of the individual algorithms.

---

> ### Author Response · Authors · 2022-09-17
> **Response to Reviewer do8M**
>
> Dear Reviewer do8M,
>
> Thank you for your detailed review! We are glad that you think our paper is a good and comprehensive exposition and provides insights into the possible motivation of existing literature.
>
> We hope that you find our work interesting despite that its contribution, impact, and significance might be modest at most.
>
> We have edited the paper according to your requests and significantly expanded it, even though this might not increase its impact:
>
> > W1) This is more on the impact side and future work, i.e. having a better/stronger framework
>
> We would love to provide a better and stronger framework. Even the current work has taken considerable effort (~200h), and we are very glad that several existing approaches fit so neatly into it. We would love any suggestions for additional directions to consider.
>
> We have added another insight: we can use the same notation ($H’’$) to both express Fisher information and observed information. The observed information is $H’’[y \mid x,\omega^*]$, and Fisher information is simply:
>
> $H’’[Y \mid x,\omega^*]=\mathbb{E}_{p(y \mid x, \omega^*)} H’’[y\mid x, \omega^*]$
>
> as an expectation over $p(y \mid x,\omega^*)$). We hope this will make reasoning about the information quantities and approximations even easier. For example, see the analogy between Table 1 and Figure 1 (both added to the new paper revision). Several results, like e.g. the GGN approximation appear trivial now: GGN is just assuming that $H’’[y \mid x, \omega^*] \approx H’’[Y \mid x,\omega^*].$
>
> > W2) Again on the impact side, but more immediate. It would be good to consider at least one new intermediate approach in detail, either to show that it might have new strengths or to motivate a negative result of why it would fail and avoid other people from wasting time on it in the future.
>
> While this might fall short of adding a fully new approach, we have added an example to §7 about Ash et al, 2021 ablating the wrong objective when comparing between trace and log determinant versions of their objective. The issue is obvious when taking into account the contributions of this paper: they end up minimizing the EIG in their ablation, which is trivially going to underperform as they essentially pick the least informative samples. They seem unaware of the chain of approximations/the origin of their objective function, which suggests a different objective to ablate with (the log determinant approximation of EPIG that we derive). We suggest this as a more interesting ablation for future work.
>
> > W3) The work is narrowly tailored in an information theoretic/bayesian perspective, but many of the comments regarding entropy and information gain approximation can also be tackled from a sub-modular function approximation perspective. It would be important to at least introduce this to reader as a potential separate path when looking for unifying frameworks.
>
> We have added a reference to submodularity to the background (§2) and expanded the sub-section on SIMILAR and PRISM in §7 to include details on the information functions used in the submodularity-based approach as they relate to the context of this paper. Submodularity is crucial for the greedy selection of acquisition batches, and we have expanded the paper to include this.
>
> > W4) Easier said than done before trying to write the paper myself. Maybe spread section 7 across Sec. 5/6 to be closer to the logical path, or include a proof graph as a visual aid to navigate the derivation.
>
> Could you take a look at the major paper revision we have uploaded and let us know if the expanded introduction and sections make it easier to navigate the paper? We are happy to add a proof graph if you think that would further improve this revision.
>
> > “Similarly, many of the original methods were already originally derived starting from an information measure perspective but had to cut corners and implement heuristics to become practically efficient presented (e.g. BADGE replacing k-dpps with k-means++). This makes fitting them in a clean framework (including the one presented by the authors) more difficult.”
>
> BADGE, for example, does not attempt to use an information-measure perspective. It only suggests using a k-DPP on a loosely motivated similarity matrix based on gradient embeddings without providing an information-theoretic perspective behind it. BAIT is motivated using Bayesian linear regression, but the paper only draws a connection to the EIG/BADGE instead of noting that it is actually computing the transductive EPIG objective. We hope we can provide some useful clarifications for readers who are reading up on different approaches and how they can be connected conceptually and practically.
>
> We apologize for the delay in responding: updating the paper took longer than expected. We hope the update is to your liking.
>
> Thank you so much for your review, and please let us know what you think,\
>  the Authors

---

### Author Response · Authors · 2022-11-06
**Thank you so much for the very helpful reviews and constructive feedback**

Dear Reviewers and Action Editor,

Thank you so much for your kind and extensive reviews and constructive feedback. We are very impressed with the level of detail provided and very happy with how much the paper has been improved by your constructive interactions and encouragement. Submitting to TMLR has been a great experience.

We have uploaded the camera-ready and made the repository public.

We have fixed a few typos for the camera-ready and made minor edits.

Thank you again and best wishes,\
the Authors

---

### Decision · Action_Editors · 2022-10-14

**Recommendation:** Accept as is

**Comment:**

The paper considers the acquisition function in active learning and active sampling. It shows how a wide array of existing acquisition functions can be understood as approximations to information gain, emerging from a second-order posterior approximation. Through their paper, the authors contribute to a more unified perspective on these different approaches and to a better understanding of the involved approximation choices.

The paper's original version was arguably hard to follow, but it significantly improved during the review due to the authors' ongoing efforts and in response to the reviewers' guidance. The authors significantly improved the paper through reorganization, the addition of clarifying figures, and the inclusion of more background material on existing approaches. It will serve as a reference for researchers working on active learning.

**Audience:**

Yes, the paper will be of interest to a broad readership interested in information theoretical aspects and foundations of active learning.

**Claims And Evidence:**

The paper draws connections between different active learning and active sampling strategies and views them through a unified information-theoretical perspective. All claims made seem accurate and supported by rigorous mathematics.